# Satellite (GOSAT-2 CAI-2) retrieval and surface (ARFINET) observations of aerosol black carbon over India

**Mukunda M. Gogoi**[1], **S. Suresh Babu**[1], **Ryoichi Imasu**[2], and **Makiko Hashimoto**[3]

[1]Space Physics Laboratory, Vikram Sarabhai Space Centre, Indian Space Research Organisation, Thiruvananthapuram 695-022, India
[2]Atmosphere and Ocean Research Institute, The University of Tokyo, Chiba 277-8568, Japan
[3]Earth Observation Research Center, Japan Aerospace Exploration Agency, Ibaraki 305-8505, Japan

**Correspondence:** Mukunda M. Gogoi (mukunda.mg@gmail.com) and Ryoichi Imasu (imasu@aori.u-tokyo.ac.jp)

**Abstract.** Light-absorbing black carbon (BC) aerosols strongly affect Earth's radiation budget and climate. This paper presents satellite retrieval of BC over India based on observations from the Cloud and Aerosol Imager-2 (CAI-2) on board the Greenhouse gases Observing Satellite-2 (GOSAT-2). To evaluate and validate the satellite retrievals, near-surface BC mass concentrations measured across the Aerosol Radiative Forcing over India NETwork (ARFINET) of aerosol observatories are used. Then the findings are extended to elucidate global BC features. The analysis reveals that this satellite retrieval clearly demonstrates the regional and seasonal features of BC over the Indian region, similarly to those recorded by surface observations. Validation and closure studies between the two datasets show RMSE $< 1$ and absolute difference below $2\,\mu g\,m^{-3}$ for $> 60\,\%$ of simultaneous observations, exhibiting good associations for December, January, and February ($R$ of approximately 0.73) and March, April, and May ($R$ approx. 0.76). Over the hotspot regions of India, satellite retrievals show a soot volume fraction of approx. 5 %, columnar single-scattering albedo of approx. 0.8, and BC column optical depth of approx. 0.1 during times of the highest BC loading, which are comparable to other in situ and satellite measurements. In terms of global spatiotemporal variation, satellite retrievals show higher BC occurring mostly in areas where biomass burning is intense. Overall, this study highlights the effectiveness of satellite retrieval of BC, which can be used effectively for the regular monitoring of BC loading attributable to vehicular, industrial, or biomass burning activities.

## 1 Introduction

The convergence of various studies using experimentation and modeling, all including the climate warming potential of atmospheric black carbon (BC), necessitates accurate quantification and seasonal source characterization of BC on regional and global scales (Bond et al., 2013; Gustafsson and Ramanathan, 2016; IPCC, 2021). Concerted efforts have been made to elucidate the radiative properties of BC (warming as well as offsetting of aerosol scattering effects) originating from the incomplete combustion of biofuel or fossil-fuel sources. Although a nearly accurate estimation of BC can be made using an in situ approach (uncertainty in BC measurements $< 5\,\%$–$10\,\%$; Manoj et al., 2019), most studies confined to in situ measurements (ground based or airborne) lack sufficient spatial coverage. Similarly, model-simulated BC has good spatiotemporal coverage subject to deviations from the real BC environment, mainly because of inaccurate model inventories and meteorological input available for simulations (Vignati et al., 2010). In this regard, retrieval of BC from satellite-based radiation measurements, synchronized with the ground-based point measurements, is

a novel method of quantifying and classifying the real BC environment across distinct geographic regions worldwide. Nevertheless, retrieving the backscattering signal accurately from optically thin BC aerosols lofted above highly heterogeneous land surfaces such as vegetated, desert, semiarid, and urban regions, having diverse surface reflectance properties, presents a daunting task. The complex optical properties of BCs caused by their highly heterogeneous sources and transformation processes add further complexity to satellite retrieval, especially over land. Several new algorithms have been developed for aerosol retrieval over land (e.g., Multi-angle Imaging Spectroradiometer (MISR) retrieval by Diner et al., 1998; the Dark Target method by Levy et al., 2007; the non-linear optimal estimation algorithm by Wurl et al., 2010; Multi-Angle Implementation of Atmospheric Correction (MAIAC) by Lyapustin et al., 2011; the Deep Blue aerosol retrieval algorithm by Hsu et al., 2013; the UV method by Fukuda et al., 2013; multi-angle and polarization measurements of radiations by Dubovik et al., 2011, 2021; the GOCI Yonsei aerosol retrieval (YAER) algorithm by Choi et al., 2016; the multi-wavelength and multi-pixel method (MWPM) by Hashimoto and Nakajima, 2017), but retrievals of BC from satellite-based radiation measurements have been few. Several attempts have been undertaken to identify dominant aerosol types using surface-based remote sensing of aerosols (e.g., Omar et al., 2005; Lee et al., 2010; Shin et al., 2019) and satellite-based remote sensing of aerosols (e.g., Higurashi and Nakajima, 2002; Kim et al., 2007; Lee et al., 2010; Kahn et al., 2015; Kim et al., 2018; Mao et al., 2019; Falah et al., 2023), but accurate quantification of the concentrations of various aerosol types from satellite remote sensing data persists as a challenge. A few recent studies are producing useful results for progress in this direction.

Based on effective medium approximations of mixture morphology and a statistically optimized aerosol inversion algorithm, Bao et al. (2019) have reported the retrieval of surface mass concentrations of BC from Polarization and Anisotropy of Reflectances for Atmospheric Sciences coupled with Observations from a Lidar (PARASOL) measurements. Their satellite retrieval strategy incorporates both internal and external mixing models of BC, with BC fractions limited to 5 %. Among the six PARASOL channels used for the retrieval process, the results obtained at 870 nm were used because BC more strongly absorbs light at this wavelength than other light-absorbing species do. Overall, this algorithm demonstrated a strong capability for detecting aerosols in polluted atmospheres. In another study reported by Bao et al. (2020), MODIS Aqua Level-1B observations (MYD021KM) at three visible infrared channels (470, 660, and 2100 nm) were used to estimate the columnar concentrations of BC aerosols based on BC and non-BC Maxwell–Garnett effective medium approximation. By incorporating wavelength-dependent refractive indexes of BC, this approach led to reliable estimation of BC. POLDER/PARA-SOL satellite observations were also used by Li et al. (2020)

to retrieve BC and brown carbon concentrations based on an aerosol component approach of Li et al. (2019). Apart from satellite observations, efforts have been made to retrieve BC from ground-based remote sensing data. Hara et al. (2018) reported the retrieval of BC from multi-wavelength Mie–Raman lidar observations, based on a modified algorithm reported by Nishizawa et al. (2017). Ceolato et al. (2022) reported a direct and remote technique to estimate the BC number and mass concentration from picosecond short-range elastic backscatter lidar observations.

This paper presents the regional distribution of BC over India based on satellite-based retrievals from Thermal And Near infrared Sensor for carbon Observation–Cloud and Aerosol Imager-2 (TANSO-CAI-2) observations made from the Greenhouse gases Observing Satellite-2 (GOSAT-2). The main purpose of CAI-2 is to derive cloud areas to improve the accuracy of greenhouse gas (GHG) retrieval from Fourier transform spectrometer (FTS) measurements in addition to ascertaining the concentrations of the BC mass and fine particulate matter ($PM_{2.5}$). The retrieval technique of BC from CAI-2 measurements is based on fine-mode aerosol optical depth (AOD) estimates at multiple pixels, along with estimation of the volume mixing ratio of BC in fine-mode particles. The AOD and aerosol absorption properties can be retrieved simultaneously using the relation of surface reflectance and observed reflection passing through the aerosol layer at multiple pixels. Using combined information from multiple wavelengths, fine-mode and coarse-mode AODs are retrieved separately. The MWPM method reported by Hashimoto and Nakajima (2017) adopts a combination of an optimal method based on Bayesian estimation and smoothing constraint to horizontal aerosol distribution to solve the problem. In contrast to conventional pixel-by-pixel methods, the MWPM method can simultaneously retrieve fine-mode and coarse-mode AODs, soot volume fraction in fine-mode aerosols, and surface reflectance over heterogeneous surfaces over multi-wavelengths and multiple pixels. Here, the soot volume fraction is assumed to be the volume mixing ratio of BC in fine-mode particles. This feature increases the accuracy of aerosol retrieval over the inhomogeneous surface, which also functions well for a homogeneous surface. Details are presented in Sect. 2.1.

To evaluate and validate the spatiotemporal distribution of BC from satellite retrieval, near-surface BC mass concentrations measured across the Aerosol Radiative Forcing over India NETwork (ARFINET; Babu et al., 2013; Gogoi et al., 2021) of aerosol observatories are used. Then the findings are extended to elucidate the global BC features. The main objective of ARFINET is the measurements of various aerosol parameters (e.g., columnar aerosol optical depth, BC mass concentrations) to characterize their heterogeneous properties in space, time, and spectral domains; develop periodic and accurate estimates of aerosol radiative forcing over India; and assess their effects on regional and global climates. Since its modest beginnings in 1985, the network has ex-

panded to more than 40 observatories today. Supplement Table S1 provides additional details related to ground-based observational locations of the ARFINET. The stations are arranged and grouped with respect to their geographic positions (Fig. 1) in the Indo-Gangetic Plains (IGPs); northeastern India (NEI); northwestern India (NWI); Himalayan, sub-Himalayan, and foothill regions (HIMs); central India (CI); peninsular India (PI); and island locations (ILs). The systematic and long-term monitoring of BC in the ARFINET began in 2000, followed by the gradual extension of its observational sites in phases. In this study, the use of ground-based BC from the ARFINET is unique in that the BC over the Indian region is highly heterogeneous spatially and temporally (Manoj et al., 2019; Gogoi et al., 2017, 2021). With rapidly growing industrial and transport sectors, mixed with diverse uses of fossil fuels and biofuels in the domestic and industrial sectors, the Indian region is a complex blend of emissions and atmospheric processes (Babu et al., 2013; Gogoi et al., 2021). Whereas the shallow atmospheric boundary layer has very high concentrations of BC near the surface in winter (December–February), especially over the northern part of India (Nair et al., 2007; Pathak et al., 2010; Gogoi et al., 2013; Vaishya et al., 2017), the synoptic circulations and convective processes are dominant in the horizontal and vertical redistribution of BC in the pre-monsoon (March–May) and monsoon (June–September) seasons (Babu et al., 2016; Nair et al., 2016; Gogoi et al., 2019, 2020). Consequently, synergistic studies of the regional BC distribution by combining satellite and surface measurements over the Indian region are extremely valuable for improving retrieval accuracy as well as expanding it to the elucidation of global BC distribution in near real time.

## 2   Data and methodologies

### 2.1   Retrieval of aerosol properties from Cloud and Aerosol Imager-2 (CAI-2)

CAI-2 on board the GOSAT-2 satellite is a push-broom imaging sensor that records backscattered radiances at seven wavelengths/10 spectral bands in the ultraviolet (UV: 339, 377 nm), visible (VIS: 441, 546, 672 nm), and near infrared (NIR: 865, 1630 nm) equipped in forward- (bands: 339, 441, 672, 865, and 1630 nm) and backward-looking (bands: 377, 546, 672, 865, and 1630 nm) directions (±20°). For cloud discrimination and for deriving aerosol properties, CAI-2 Level-1B (L1B) data are used. These include spectral radiance data per pixel converted from sensor output (Hashimoto and Shi, 2020).

A flowchart of the CAI-2 L2 pre-processing algorithm is presented in the Supplement in Fig. S1. Radiances measured at forward-viewing bands (3–5) and backward-viewing bands (8–10) are used for cloud discrimination. The cloud detection algorithm (Ishida and Nakajima, 2009; Ishida et al., 2018) uses reflectance (at the top of the atmosphere) of these

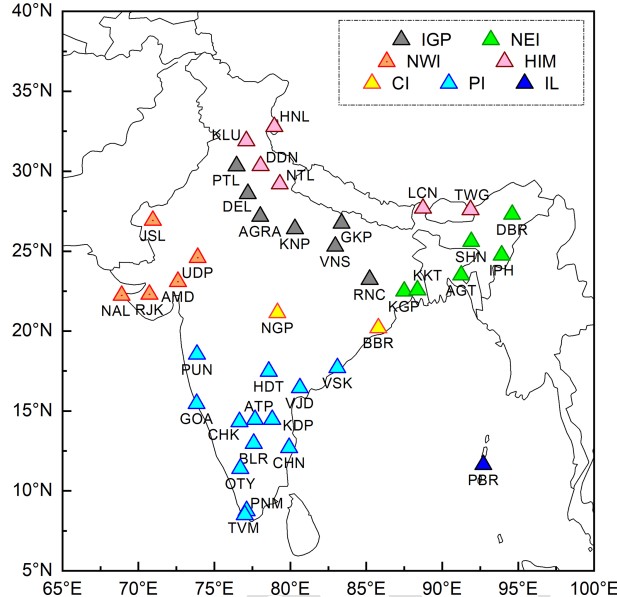

**Figure 1.** Network of aerosol observatories over India, distributed in the Indo-Gangetic Plains (IGPs); northeastern India (NEI); northwestern India (NWI); Himalayan, sub-Himalayan, and foothill regions (HIMs); central India (CI); peninsular India (PI); and island locations (ILs). More details about ground-based observational locations in the ARFINET are provided in Table S1.

bands for detecting clouds from 11 recurrences (1 month before and after the observation date) (Oishi et al., 2020). A flowchart of the Cloud and Aerosol Unbiased Decision Intellectual Algorithm (CLAUDIA3; Ishida et al., 2018; Oishi et al., 2017) used for cloud screening of GOSAT-2 CAI-2 data is given in Fig. S2. CLAUDIA3 is designed to find the optimized boundary between clear and cloudy areas automatically based on a supervised pattern recognition that uses support vector machines (SVMs; Oishi et al., 2017). Before using the radiance (L1B) data in CLAUDIA3, pre-processing is done to discriminate between day and night, saturation flags, missing flags, polar regions, water and land areas, and sunglint areas for water areas except for polar regions. Subsequently, solar reflection properties by clouds and ground surface are examined. These include (i) solar reflectance and reflectance ratio in VIS and short-wave infrared (SWIR) regions, (ii) wavelength dependence of reflectance in the VIS and NIR region, (iii) normalized difference vegetation index (NDVI) test for cloud discrimination over vegetated areas, and (iv) reflectance ratios between NIR and SWIR bands for cloud discrimination over desert areas (details in cloud discrimination processing ATBD). Following this, the CLAUDIA3 algorithm performs cloud discrimination by SVMs (Ishida et al., 2018) to ascertain thresholds using multivariate analysis objectively. SVM is a supervised pattern recognition method, which first determines a decision function (called separating hyperplane) that defines clear or cloudy condi-

tions according to the features of training samples (support vectors) in combination with a decision function.

The next step after cloud discrimination is cloud shadow detection. A minimum reflectance criterion is used for this purpose (Fukuda et al., 2013), which incorporates the difference between the first and second minimum reflectance at UV (339 nm in forward-viewing band 1 and 377 nm in backward-viewing band 6), visible (672 nm in forward-viewing band 3 and backward-viewing band 8), and NIR (865 nm in forward-viewing band 4 and backward-viewing band 9) bands. The first and second minimum reflectances at 672 nm are selected from multiple days from about 2 months of data between $X_{day} - n_1$ and $X_{day} + n_2$ day, where $X_{day}$ is an analysis day and $n_1$ and $n_2$ respectively represent the number of scenes required before and after the analysis date that take the same path as the analysis date. When the difference between the first and second minimum is smaller than a threshold for band 1 (339 nm, forward viewing) and band 6 (377 nm, backward viewing), i.e., $R_{(second, min)band 1,6} - R_{(first, min)band 1,6} < 0.10$, and greater than a threshold for band 4 (865 nm, forward viewing) and band 9 (865 nm, backward viewing), i.e., $R_{(second, min)band 4,9} - R_{(first, min)band 4,9} > 0.06$, the first minimum reflectances of bands 3 and 8 are judged to be affected by cloud shadows, and the second minimum reflectance is selected as the minimum reflectance (Fukuda et al., 2013). The advantage of using near-UV wavelengths is that the surface reflectance at UV over land is less than that at visible wavelengths, as already applied for aerosol retrieval in Total Ozone Mapping Spectrometer and Ozone Monitoring Instrument (TOMS and OMI; Torres et al., 1998, 2002, 2007, 2013), and MODIS (Hsu et al., 2004, 2006).

After cloud and cloud shadow correction, the influence of atmospheric molecular scattering (Rayleigh scattering) is corrected from the minimum reflectance data. For this correction, radiative transfer calculations are performed in advance and look-up tables (LUTs) are generated for atmospheric single- and multiple-scattering components of reflectance, unidirectional transmittance, and spherical albedo. Based on this, the effect of atmospheric molecular scattering is removed from the minimum reflectance data for different combinations of satellite–solar geometry. The surface albedo ($A_g$) is estimated from the atmospherically corrected minimum reflectance data using the following equations:

$$A_g = \frac{1}{C + r_{Band(i)}(\tau)}, \tag{1}$$

$$C = \frac{t_{Band(i)}(\tau, \mu_o)\, t_{Band(i)}(\tau, \mu_1)}{R_{Band(i)}(\mu_1, \mu_o, \varphi)/T^2_{gas,Band(i)} - R_{Atmos(i)}(\mu_1, \mu_o, \varphi)}. \tag{2}$$

In those equations, $\mu_1$, $\mu_o$, and $\varphi$ respectively denote satellite zenith angle, solar zenith angle, and relative azimuth angle. $R$ and $T_{gas}$ respectively denote the apparent reflectance and transmission of light-absorbing gas. Subscript $i$ denotes observation band numbers from 1 to 10. $R_{atmos} = R_{single} +$

$R_{multiple}$. $\tau$ stands for the optical thickness of the atmosphere, $t(\tau, \mu_o)$ and $t(\tau, \mu_1)$ are unidirectional transmittance, and $r(\tau)$ is spherical albedo. The parameters $t$, $r$, and $T_{gas}$ are obtained by LUTs (details in GOSAT-2 TANSO-CAI-2 L2 pre-processing ATBD).

### 2.1.1   Retrieval of AOD and SSA

For retrieval of columnar aerosol optical depth (AOD) and aerosol single-scattering albedo (SSA) from the satellite-received path radiances, the MWPM inversion algorithm (Hashimoto and Nakajima, 2017) is used. This algorithm uses information contained in different pixels with different surface reflectance, and it is assumed that aerosol properties vary slowly or almost negligibly in the horizontal direction (over different pixels) where the variations in surface properties are significant. Consequently, the variations in the upward radiances over different pixels are assumed to be varying because of variations in the surface reflectance at the respective pixels. Under this assumption, when there is an increasing aerosol load over all the pixels under consideration, the satellite-reaching upward (backscattered) radiance increases over a dark surface. In comparison to that, the change in the magnitude of upward radiance with increasing aerosol load over brighter surface reflectance is less. Because as the surface reflectance increases, the absorption of light in the atmosphere and the backscattering of radiance to the surface increase, which results in a decrease in the net upward radiance. At some specific surface reflectance, the net upward radiance does not change with increasing aerosol load in the atmosphere because the increasing absorption and backscattering of light caused by the aerosol load in the atmosphere fully compensate for the increasing surface reflectance, leaving net-zero upward radiance. Surface reflectance of this kind is designated as neutral reflectance where the apparent reflectance is equal to the surface reflectance. The difference between the apparent reflectance and surface reflectance is the net reflectance. For surface reflectance beyond the neutral reflectance, the surface reflectance is predominant over the apparent reflectance, resulting in a darkening effect of the atmosphere on the surface (Kaufman et al., 1987). It is noteworthy that the balance between the brightening of the surface by atmospheric scattering and darkening by aerosol absorption (i.e., critical surface reflectance or neutral reflectance) varies with the values of SSA. Each value of SSA has a corresponding value of neutral or critical reflectance, for which the upward radiance is almost independent of the AOD.

The above methodology, which was adapted by Hashimoto and Nakajima (2017), is an extension of the method reported by Kaufman (1987). However, the methodology uses information of aerosol and surface properties at multiple wavelengths and multiple pixels of satellite image. Because the variation in radiances takes place with variation in AOD depending on aerosol light

scattering (or single-scattering albedo – SSA) and surface reflectance, this principle is suitable for successful retrieval of SSA over different surface reflectance areas. Considering that no change occurs in the measured radiances between a clear (low AOD) and a hazy (high AOD) day, the critical reflectance is determined from satellite radiances. The spatially distributed critical surface reflectance is then used to derive AOD and SSA over multiple pixels using a theoretical relation between critical reflectance, AOD, and SSA, computed for a given aerosol scattering phase function. Radiative transfer equations (RTEs) are solved together for the information contained in radiances at each of the pixels with different surface reflectance (Hashimoto and Nakajima, 2017). The simultaneous use of short and long wavelengths in the CAI-2 bands is very effective for aerosol retrieval when the surface is covered by vegetation and bare soil, depending on the location.

The inversion method developed based on the concept above (Hashimoto and Nakajima, 2017) is a combination of the maximum a posteriori optimal method (Rodgers, 2000) and a special formulation of the GRASP method (Dubovik et al., 2011, 2021). The inversion analysis is conducted over different sub-domains, where the retrieved values of the optimal set of AOD, SSA, and surface reflectance at one domain are regarded as Dirichlet boundary conditions for the next domain.

### 2.1.2 Uncertainty in AOD and SSA retrieval

The uncertainty in the retrieval of AOD using the MWPM inversion algorithm over heterogeneous surfaces is found to be within $\pm 0.062$, $\pm 0.048$, and $\pm 0.077$ for $AOD500_{fine}$, $AOD500_{coarse}$, and $AOD500_{total}$ respectively (Hashimoto and Nakajima, 2017). These results are based on the comparison of AOD retrieval from CAI measurements of radiances with AOD data obtained from AERONET (Holben et al., 1998) and SKYNET (Nakajima et al., 2007). Comparison of the CAI-retrieved SSA (at 674 nm) with that of the AERONET-observed values (SSA at 675 nm) revealed the retrieval accuracy of SSA within 0.05. Over the homogeneous surface, the random measurement error of the retrieval parameters is below 2 %.

### 2.1.3 Deriving BC mass concentration

The BC mass concentration ($M_{BC}$) is derived (National Institute for Environmental Studies, GOSAT-2 Project, 2021) using the size distribution of fine-mode particles, the fine-mode AOD at 550 nm ($\tau 550_{fine}$), and the volume fraction of BC in fine-mode particles ($f_{BC}$). The expression for $M_{BC}$ can be given as shown below.

$$M_{BC} = \frac{1}{m} f_{BC} \rho_{BC} \int_{r_{min}}^{r_{max}} \frac{dV_{fine}(\tau 550_{fine})}{d\ln r} d\ln r \qquad (3)$$

In the above equation, $\rho_{BC}$ denotes the density of BC (approx. $1.8\,\mathrm{g\,cm^{-3}}$), $V_{fine}$ stands for the volume of fine-mode particles, $r$ denotes the radius of particles, and $m$ is the aerosol height information parameter (approx. 1000 m for this study). As $M_{BC}$ expresses 1000 m averaged values of column fine-mode aerosol particle amount, the definition differs from BC mass concentrations obtained using in situ ground-based measurements.

For estimation of $f_{BC}$, an internal mixture of fine-mode aerosols (composed of 75 % sulfuric acid particles and soot; mode radius $\sim 0.175\,\mu m$ and logarithmic dispersion of the lognormal volume size distribution $\sim 0.8$) and the volume fraction of soot particles (indicated as soot volume fraction, SF) are considered representative of aerosol light absorption by the fine-mode particles. Thus, $f_{BC} = V_{soot}/V_{fine}$, where $V_{soot}$ denotes the soot volume in the fine mode only. In the beginning, the a priori value of soot is assumed to be 0.01, and the retrieval parameter $u$ is investigated based on its a priori state $u_a$. Several a priori values around the true state $u_t$ are considered in the experiment, such as $u_t \pm 1.0u_t$ for $AOT_{500fine}$, $AOT_{500coarse}$, and SF and $u_t \pm 0.01u_t$ for surface reflectance. The a priori values of $AOD_{500fine}$ and $AOD_{500coarse}$ are considered to be 0.2. Iteration in the solution search is stopped when the threshold is $< 0.02$.

In this simple approximation, various other mixing states of aerosols such as half internal and half external, core shell, and aggregated ones (Hashimoto et al., 2017, and references therein) are ignored. Consequently, SF should be regarded as an equivalent value of soot in fine-mode particles, where the absorption property of aerosol is attributed only to the BC particles in the fine-mode regime. Because the BC mass distribution shows a mode of 100–300 nm (Kompalli et al., 2021), having stronger absorption in the NIR region, light absorption by BC is significant mostly in the fine-mode regime. Light absorption by other light-absorbing aerosols such as brown carbon and dust (coarse particles) responds strongly to radiation perturbation in the near-UV region (Mahowald et al., 2014). For the wavelength dependence of light absorption by BC, the complex refractive index of soot particles (d'Almeida et al., 1991) is considered in the retrieval process. However, aerosol light absorption in the coarse-mode domain is not considered in this assumption.

To understand the uncertainty in satellite-received radiances because of different mixing states of aerosols having varying BC and dust fractions, a sensitivity study using 6S radiative transfer code (Vermote et al., 1997) was conducted. The 6S code is used widely for simulating satellite-reaching radiation under different combinations of sun–satellite geometry and aerosol loads in the atmosphere. In the present simulations, the surface is considered homogeneous Lambertian. It can be observed (Fig. S3) that the BC fraction (at 880 nm) is significantly more sensitive to satellite-reaching radiation under higher aerosol loadings (AOD > 0.5) and under higher-surface-reflectance conditions; no marginal distinction can be made between BC and non-BC conditions

under AOD < 0.5. Variations in satellite-reaching radiation are less than 5 % for 1 % dust in the aerosol mixture and for BC fractions varying between 1 % to 10 % under low-aerosol-loading conditions (AOD of approx. 0.1). On the other hand, 10 % variation in BC fraction in the aerosol mixture with dust fractions varying between 1 % to 10 % change the apparent reflectance by approx. 10 % under heavy aerosol loading (AOD of approx. 2.0) and higher-surface-reflectance ($\sim 0.5$) conditions. This exercise demonstrates that ignoring dust contributions in the aerosol mixture engenders less uncertainty in satellite retrieval of BC. The retrieval uncertainty is lower over brighter surfaces when the aerosol load is high. Overall, one can note that consideration of the accurate mixing state (internal and external) of aerosols is important for the accurate computation of the effective refractive index and size distribution of aerosols. Lesins et al. (2002) reported that the optical properties of the internal mixture of BC and ammonium sulfate can differ by as much as 25 % (for the dry case) and 50 % (for the wet case) from those of the external mixture.

With the aforementioned uncertainties, the sensitivity study indicates that SF is underestimated under low-aerosol-loading conditions (AOD < 0.2) over highly reflective surfaces. This is because the retrieval uncertainty in AOD is higher over the high-reflectance surface, which engenders the overestimation of $AOD_{500fine}$. For higher-aerosol-loading conditions ($AOT_{500total} > 0.4$), the MWPM algorithm simultaneously determines $AOD500_{fine}$, $AOD500_{coarse}$, and SF respectively within errors of $\pm 0.06$, $\pm 0.05$, and $\pm 0.05$.

## 2.2 Estimation of BC column optical depth

By using the values of the soot volume fraction ($f_{BC}$) along with the mass absorption efficiency of BC, BC columnar optical depth ($BC_{AOD}$) is estimated. As demonstrated by Wang et al. (2013), the expression for $BC_{AOD}$ can be given as

$$BC_{AOD} = \sigma_{abs}\rho_{BC}V_{BC}, \tag{4}$$

where $\sigma_{abs}$ represents the BC mass absorption efficiency (MAE), $\rho_{BC}$ is the density of BC (assumed as $1.8\,\mathrm{g\,cm^{-3}}$), $V_{BC}(= f_{BC}.V_{total})$ is the volume concentration of BC in the vertical column, and $V_{total}$ is the total volume concentration of aerosols in the vertical column. Following Schuster et al. (2005), the volume concentrations of BC can be estimated from the columnar mass concentrations of $BC_{col}$ (in $\mu\mathrm{g\,m^{-2}}$, up to 1 km altitude in this study) as given below.

$$BC_{col} = f_{BC}\rho_{BC}\int \frac{dV}{d\ln r}d\ln r \tag{5}$$

Assuming that MAEs do not change vertically, a constant value of $MAE = 10\,\mathrm{m^2\,g^{-1}}$ is assumed (Kondo et al., 2009). The BC mass absorption efficiency (i.e., absorption coefficients of the particles divided by the mass concentrations of BC in the aerosol) shows light-absorbing efficiency of a certain amount of BC having different mixing and sizes (Martins

et al., 1998). Several investigators have reported the MAE of BC varying as $4.3$–$15\,\mathrm{m^2\,g^{-1}}$, even though the measured values for freshly generated BC fall within a narrow range of $7.5 \pm 1.2\,\mathrm{m^2\,g^{-1}}$ at 550 nm (Bond et al., 2013). Sand et al. (2021) also reported a model mean value of MAE as 10.1 (3.1 to 17.7) $\mathrm{m^2\,g^{-1}}$ (550 nm).

## 2.3 Surface BC measurements in the ARFINET

Near-surface mass concentrations of BC were obtained from the multi-wavelength Aethalometer measurements in the ARFINET. The Aethalometer measures the rate of increase in optical absorption due to BC deposited on a filter spot (Hansen et al., 1984). By knowing the change in optical attenuation by the volume of air (i.e., the mass flow rate multiplied by the sampling time) and the spot area of the filter, the BC concentrations (in $\mu\mathrm{g\,m^{-3}}$) can be estimated. Measurement of the rate of change in optical absorption on a single collecting spot can be subject to non-linearity because of the nature and composition of the aerosol (Park et al., 2010), which is prominent in earlier-model aethalometers (models AE-16, AE-21, AE-22, AE-31, AE-42-2, and AE-42-7) but not in the latest model (AE-33). As the spot gradually becomes darker, the calculated output concentration can be underreported, reverting to the correct value when the tape advances to a fresh spot. Assuming the existence of a continuous data record exists that spans several tape advances due to loaded and fresh tape spots, it is possible to post-process the BC data. This recalculates the BC data for each wavelength, in addition to providing the value of the filter loading compensation parameter, which is found to be indicative of aerosol properties (Drinovec et al., 2015). For this study, the BC data quality is ensured following the uniformity of measurements by aethalometers of different models. Regular servicing, calibration, and inter-comparison of the instrument output are also conducted in the ARFINET for quality assessment of collected data. The overall uncertainty in BC mass measured using the Aethalometer is estimated at about 10 %. Additional details are available from reports by Gogoi et al. (2017).

## 2.4 Fire radiative power

To understand the spatiotemporal distribution of BC related to the occurrences of biomass burning events across the globe, MODIS Collection 6 Active Fire products (MCD14ML), described herein as fire radiative power (FRP) and fire types, are used for this study. The MCD14ML (global fire location products) data include the geographic coordinates of individual fire pixels from both Terra and Aqua satellites. The FRP or fire radiative energy (FRE) is the emitted radiant energy released during biomass combustion episodes. It is a suitable parameter to ascertain the biomass combustion rates and the rate of production of atmospheric pollutants. The detailed principle underlying the

remote determination of FRP products used for this study is available in Wooster et al. (2003). This technique, called the mid-infrared radiance (MIR) method, uses data from the MIR spectral channel to estimate the FRP. The principle underpinning this technique is that the ratio of the total power emitted over the entire MIR wavelength range to the power emitted at 4 μm is approximately constant within a temperature range (approx. 600–1500 K) that is appropriate to most wildfires. Following this, the MIR radiance $L_{\mathrm{MIR,h}}$ of a fire hotspot pixel containing $n$ sub-pixel thermal components is expressed as presented below.

$$L_{\mathrm{MIR,h}} = a\varepsilon_{\mathbf{MIR}} \sum_{i=1}^{n} A_n T_n^4 \tag{6}$$

Therein, $\varepsilon_{\mathbf{MIR}}$ denotes the surface spectral emissivity in the appropriate MIR spectral band, $A_n$ represents the fractional area of the $n$th surface thermal component within the individual ground pixel, and $T_n$ stands for the temperature (K) of the $n$th thermal component. The constant $a$ (W m$^{-4}$ sr$^{-1}$ μm$^{-1}$ K$^{-4}$) is determined from empirical best fitting for the relation between blackbody temperature and emitted spectral radiance at a single wavelength. The equation above, when combined with the spectral radiance $L(\lambda)$ emitted by a blackbody at wavelength $\lambda$, relates FRP to the MIR spectral radiance of the hot pixel.

$$\mathrm{FRP_{MIR}} = \frac{A_{\mathrm{sampl}}\sigma\varepsilon}{a\varepsilon_{\mathrm{MIR}}} L_{\mathrm{MIR,h}} \tag{7}$$

In that equation, $A_{\mathrm{sampl}}$ expresses the ground sampling area (m$^2$), and $\sigma$ is Stefan–Boltzmann's constant. With $A_{\mathrm{sampl}} = 1.0 \times 10^6$ m$^2$, the FRPs for MODIS pixels are derived as presented below.

$$\mathrm{FRP_{MIR}} = 1.89 \times 10^7 (L_{\mathrm{MIR}} - L_{\mathrm{MIR,bg}}) \tag{8}$$

Here, $L_{\mathrm{MIR,bg}}$ represents the background MIR radiance estimated from neighboring non-fire ambient pixels. All radiances are in units of W m$^{-2}$ sr$^{-1}$ μm$^{-1}$ and FRP in units of Js$^{-1}$ or watts.

## 3 Results and discussion

### 3.1 Regional distribution of BC over India

GOSAT-2 makes 89 laps for observing the whole globe in 6 d (swath $\sim$920 km). Starting from the ascending node, data of each satellite revolution are defined as a CAI-2 scene. Each scene is divided into 36 equal parts (each designated as a frame) by the argument of latitude at the observation point of the central pixel. A file unit of CAI-2 archives the data of one frame. Because the scene for CAI-2 archives the data of only the day side, 18 files are generated from one satellite revolution. For the present study, data from individual files are analyzed to estimate daily

and monthly average values. For distinct geographical regions of India with a variety of emissions and transformation processes of carbonaceous aerosols, the spatiotemporal distributions of BC from satellite (GOSAT-2 CAI-2) retrieval (of the years 2019 and 2020) and surface measurements (climatological data) in the ARFINET are presented in Fig. 2 (December–January–February, DJF), Fig. 3 (March–April–May, MAM), and Fig. 4 (June–July–August, JJA), respectively representing three distinct periods of winter, premonsoon, and monsoon.

In winter (DJF), the surface observations (Fig. 2) depict the highest BC mass concentrations ($M_{\mathrm{BC}}$) in the IGP ($\sim 13.67 \pm 5.65$ μg m$^{-3}$) followed by NEI ($\sim 12.35 \pm 4.87$ μg m$^{-3}$), with $M_{\mathrm{BC}}$ exceeding 7 μg m$^{-3}$ in most locations. Several polluted locations exhibit values above 15 μg m$^{-3}$, with the highest values occurring in urban centers. BC concentrations remain lower ($< 5.5$ μg m$^{-3}$) over the NWI ($\sim 4.67 \pm 3.48$ μg m$^{-3}$), CI ($\sim 5.36 \pm 0.80$ μg m$^{-3}$), and PI ($\sim 4.51 \pm 3.02$ μg m$^{-3}$) and the lowest across the HIM (including sub-Himalayan and foothill sites; average BC $\sim 2.26 \pm 1.75$ μg m$^{-3}$). Similarly to surface observations, satellite retrievals also show higher values of BC over the IGP and NEI, with magnitudes comparable to those of the surface BC measurements. Pockets of higher BC are also apparent at some locations of PI from both satellite retrievals and surface measurements. It is also consistent with surface observations that satellite-retrieved BC is higher over the eastern coast of India. However, it is noteworthy that the intra-seasonal variation in the case of satellite retrieval is very significant, whereas near-surface measurements of BC at the point locations of the ARFINET show nearly consistent values for different months of winter.

In the pre-monsoon period (MAM, Fig. 3), the surface measurements show a gradual decline in BC from that of DJF, with a 50 %–60 % decline in seasonal average surface BC concentrations at the hotspots of IGP ($\sim 7.05 \pm 1.78$ μg m$^{-3}$) and NEI ($\sim 4.88 \pm 1.13$ μg m$^{-3}$). The intraseasonal variation in BC at different point locations of the ARFINET is also apparent during this period, with values of BC decreasing towards May. In line with this finding, the satellite retrievals also clearly show a gradual decline in BC from DJF to MAM, while retaining the consistent features of the regional hotspots of BC over the IGP and NEI, as is apparent in the surface measurements. The intra-seasonal variation in the satellite retrievals resembles that of the surface observations. Moreover, in both satellite retrievals and surface measurements, BC remains below 3 μg m$^{-3}$ over the NWI, CI, and PI regions.

In the monsoon period (JJA, Fig. 4), the surface BC concentrations decrease considerably at most locations of ARFINET. The average values of the surface-measured BC over the IGP and NEI are $3.93 \pm 1.64$ and $2.64 \pm 1.30$ μg m$^{-3}$ respectively, with $M_{\mathrm{BC}} < 2$ μg m$^{-3}$ over the rest of the regions. Resembling this, the satellite retrievals also show a decline in BC from MAM to JJA over the IGP and NEI. How-

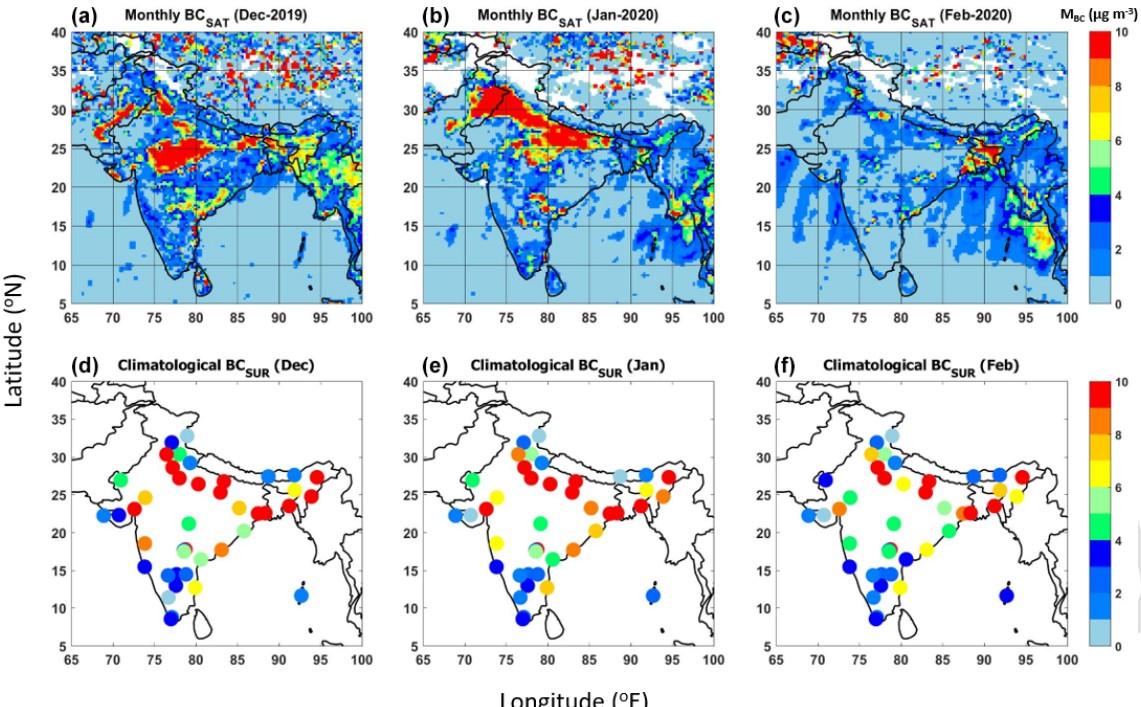

**Figure 2.** Regional distribution of monthly average BC over the Indian region from satellite (2019–2020) and surface measurements (climatological monthly average) during December–January–February (DJF), representing winter. The satellite-retrieved BCs (BC$_{SAT}$) in **(a)**–**(c)** are shown at $0.25 \times 0.25°$ spatial resolution. The surface BCs (BC$_{SUR}$) in **(d)**–**(f)** are climatological monthly average values at the point locations of the ARFINET. A minimum of 3 to more than 10 years of data are included for the estimation of the climatological average. The color bars indicate the magnitudes of monthly average BC mass concentrations.

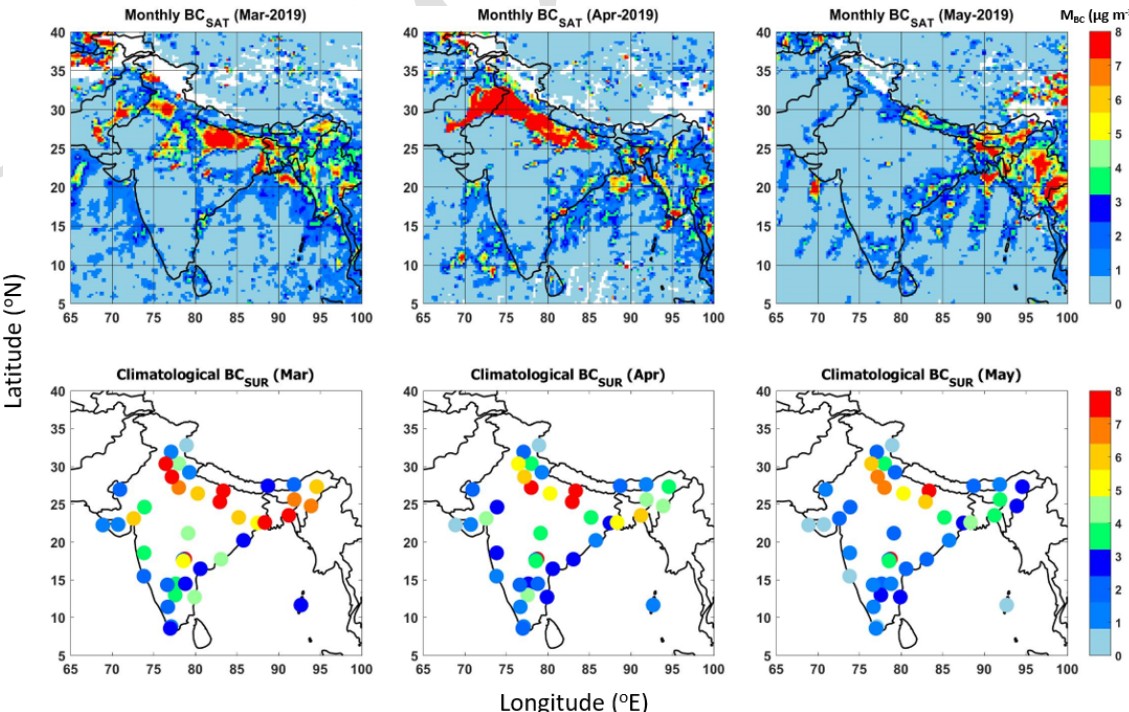

**Figure 3.** The same as Fig. 2, for March–April–May (MAM), representing the pre-monsoon.

ever, as opposed to surface measurements, satellite retrievals show higher BC ($> 3\,\mu g\,m^{-3}$) in several pockets of CI and PI regions, particularly during July and August, with values higher than those during MAM.

Based on the observations described above, the spatiotemporal distribution of BC as obtained from satellite retrievals apparently has better similarity with the surface-measured BC over the Indian region during DJF and MAM. The increase in temperature caused by increased solar heating during MAM and JJA gives rise to strong thermal convection over the Indian region (especially in the northern part), which leads to dilutions of near-surface aerosol concentrations. Depending upon the geographic position and local meteorological conditions, the strength of convection varies among locations. Because the satellite-retrieved BC is 1 km column-averaged BC concentrations, the variation in the vertical distribution of BC might lead to variable associations between satellite-retrieved and surface-measured BC concentrations for distinct geographic locations of India. Additional details on these aspects are discussed in the following sections. Apart from the vertical heterogeneity, various other factors that might lead to a discrepancy in the satellite retrieval of BC include the bias caused by the cloud-screening algorithm, especially during JJA when the cloud cover over the Indian region is extensive. Moreover, CLAUDIA3 is unable to detect optically thin clouds. A lack of accurate detection of cloud shadows can also cause overestimation in the retrieved values of aerosol parameters from CAI-2 measurements. Since the revisiting time of CAI-2 is long (6 d), the minimum reflection criterion based on the consideration of 2 months of data (1 month prior and after the measurement day) can lead to a large uncertainty in cloud shadow detection, hence the accurate estimation of minimum surface reflectance. Subsequently, these errors can propagate and add uncertainty, which can hinder the accurate estimation of aerosol parameters from CAI-2 measurements.

### 3.1.1 Satellite retrievals versus climatological surface BC concentrations

Comparison of the $1 \times 1$ degree area average BC (around each observational site) from satellite retrievals with the climatological (2015–2019) monthly average surface BC concentrations (obtained during 13:00 to 14:00 local time) at respective sites at different months of winter, pre-monsoon, and monsoon periods illustrates the consistency of satellite retrievals (Fig. S4: statistical fit parameters are given in Table S2). Linear correlation between the two datasets is the highest in May ($R \sim 0.79$); during February–August $R > 0.6$, and in December and January $R < 0.5$. In seasonal terms (Fig. S5), the satellite retrievals and surface observations show better agreement during MAM ($R \sim 0.70$). In JJA, correlation between the two datasets is weak ($R \sim 0.50$) and the lowest in DJF ($R \sim 0.43$), which indicates that satellite retrievals and surface observations show good agreement at the regional

hotspots of BC over India during winter and pre-monsoon months, but even so, a lack of consistency exists between the two datasets in winter at some of the other ARFINET observational sites.

The discrepancies between satellite retrievals and ground-based observations can be attributed to the varying roles of geographical features and to the heterogeneity of BC abundance and their vertical distribution in the atmosphere during different seasons. As the satellite-retrieved BC data are 1 km column-averaged fine-mode particle concentrations, the role of planetary boundary layer (PBL) dynamics and columnar patterns of BC distribution are crucially important for understanding the association between satellite-retrieved and surface-measured BC. In locations having a PBL height of $\sim 1\,\mathrm{km}$, better associations are expected between the two than in locations with much a extended ($> 1\,\mathrm{km}$) or shallow ($< 1\,\mathrm{km}$) PBL. Consequently, the spatiotemporal variability of the PBL is anticipated as an important factor explaining the association between satellite retrieval and climatological surface BC measurements.

The regional average BC over the entire Indian region (Fig. 5) indicates that the satellite-retrieved BC differs from surface-measured BC by $< 33\,\%$ in most months, except in July and August ($> 50\,\%$). During February–August, the magnitudes of satellite-retrieved BC are lower (underestimates, by as much as 32.6 % in February) compared to surface measurements, whereas the opposite (overestimates) is true in December–January and June–August, with the highest overestimation in August ($\sim 69\,\%$). Seasonally, the difference between the two datasets over the entire Indian region is $< 20\,\%$ in DJF and MAM and $\sim 53.5\,\%$ in JJA (Table 1). Generally speaking, the surface measurements of BC concentrations over the entire Indian region show a gradual decline from their highest values in DJF ($2.54 \pm 0.11\,\mu g\,m^{-3}$) through MAM ($2.06 \pm 0.47$) to their lowest value in JJA ($1.11 \pm 0.17\,\mu g\,m^{-3}$). Similarly, the 1 km column-averaged satellite-retrieved BC also shows the highest BC concentrations over the collocated locations of India during DJF and their gradual decline in MAM. However, the satellite-retrieved BC is found to be higher in JJA than in MAM, as opposed to the pattern seen in the case of surface-measured BC. These observations hint again at the discrepancy between satellite retrievals and surface-measured BC in JJA, whereas their absolute magnitudes and regional distributions are nearly consistent during DJF and MAM in most locations.

### 3.1.2 Satellite retrievals versus daily surface BC concentrations

After studying the regional distribution and the association between satellite-retrieved BC and climatological monthly average surface BC levels in DJF, MAM, and JJA, we examine simultaneous day-to-day values of BC from satellite retrievals and surface measurements. Here, the measured BC

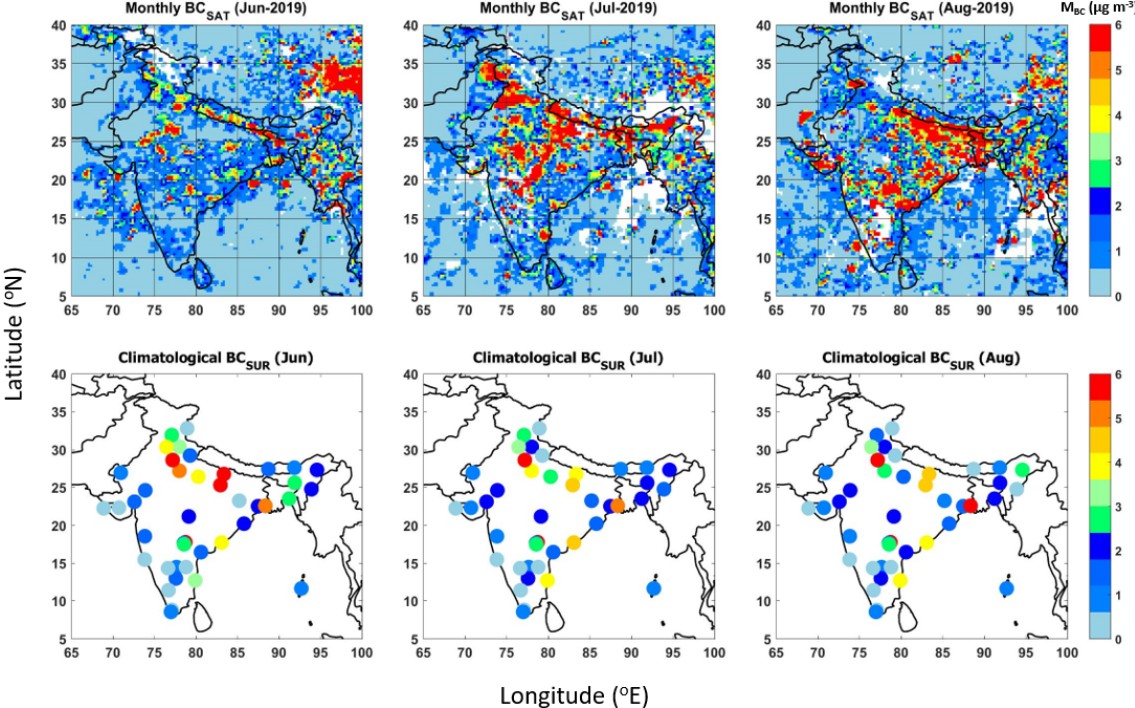

**Figure 4.** The same as Figs. 2 and 3 above, for June–July–August (JJA), representing the monsoon season.

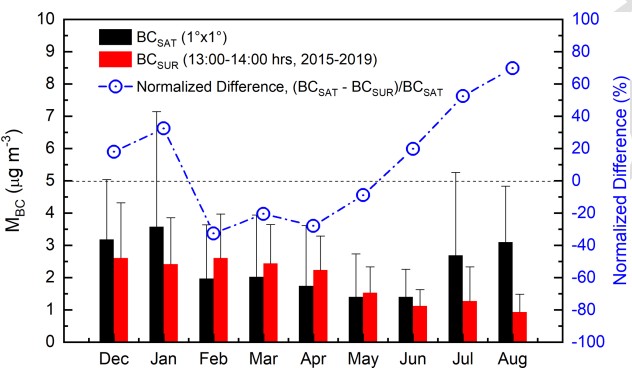

**Figure 5.** Monthly variation in the regional average values (averaged over all the locations considered for comparison) of BC concentrations from satellite retrievals ($BC_{SAT}$) and surface measurements ($BC_{SUR}$), along with the normalized difference (in %) between the two datasets.

**Table 1.** Regional average BC over India from satellite and surface measurements. The satellite-based estimate is made from $1 \times 1°$ area average values around each of the ARFINET sites, whereas the climatological surface BC is for 2015–2019 (13:00 to 14:00 local time).

| Period | Average BC over India ($\mu g\,m^{-3}$) | | |
|--------|-----------|-----------|-----------|
| | $BC_{SAT}$ | $BC_{SUR}$ | Normalized difference (%) |
| DJF | $2.91 \pm 0.84$ | $2.54 \pm 0.11$ | 12.7 |
| MAM | $1.72 \pm 0.31$ | $2.06 \pm 0.47$ | $-19.7$ |
| JJA | $2.39 \pm 0.89$ | $1.11 \pm 0.17$ | 53.5 |

concentrations in the surface are normalized to a PBL height of 1 km for use in the validation experiment. It is assumed that BC possesses a uniform vertical profile within the well-mixed PBL and that its concentrations are negligible above the PBL. Consequently, the expression relating the 1 km column concentration of BC ($BC_{SUR-N}$) and the actual BC concentrations measured at the surface within the PBL ($BC_{SUR}$) can be given as presented below.

$$BC_{SUR-N} = BC_{SUR}/h \tag{9}$$

In that equation, $h$ signifies the PBL height. The measured concentrations of BC in the PBL are assumed to be the sum of concentrations in each layer of thickness $dh$ from the surface to the PBL height $h$ (i.e., $BC_{SUR-N} = \int_0^h BC_i(h)\,dh$, where $i$ represents the number of layers from 0 to $h$); for $h = 1$, $BC_{SUR-N} = BC_{SUR}$. As the PBL height exceeds 1 km, the measured BC concentrations in the surface become greater than those measured within 1 km and vice versa. The seasonally varying PBL heights at different ARFINET sites might play an important role in elucidating the association between the satellite retrieval and the surface-measured BC. For that reason, the normalized values of surface BC concentrations ($BC_{SUR-N}$) are used in this section to evaluate and validate the simultaneous (corresponding to satellite

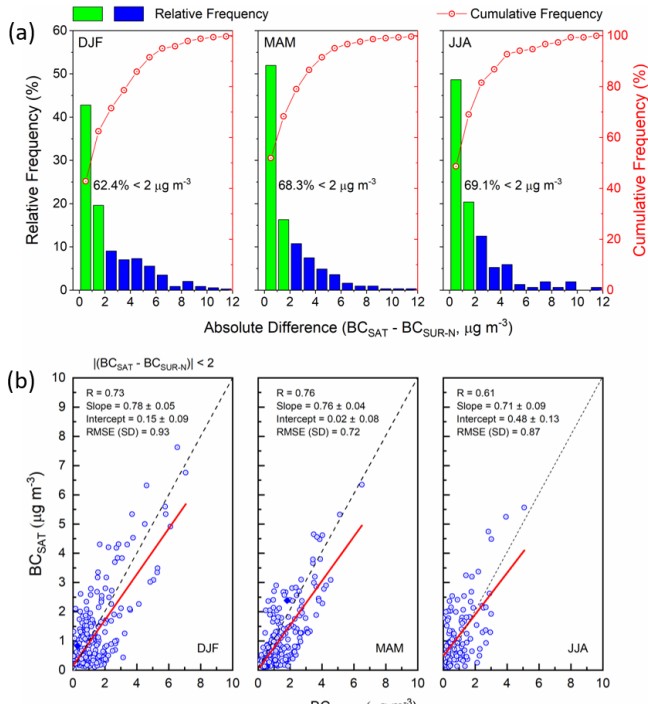

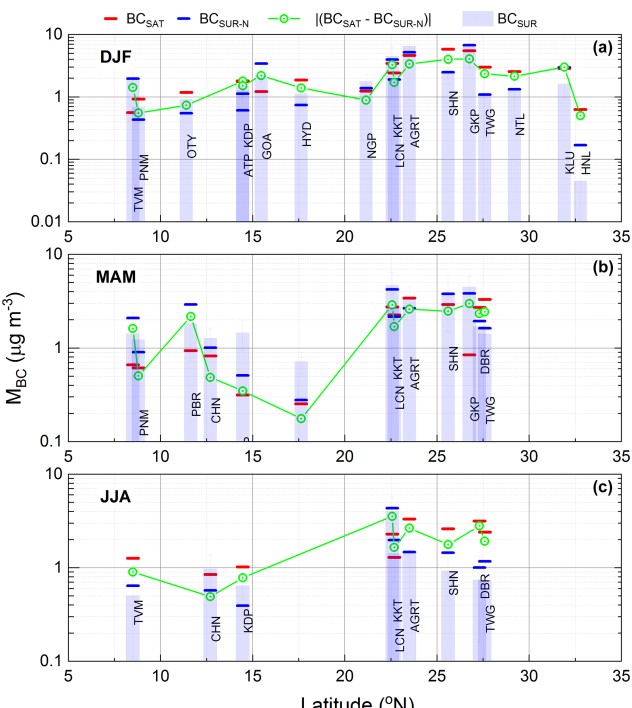

**Figure 6. (a)** Frequency counts (in percentage) of the absolute difference in BC (in $\mu g\,m^{-3}$) between simultaneous satellite ($BC_{SAT}$, averaged over $1 \times 1°$ area around each of the ARFINET sites) and normalized surface BC ($BC_{SUR-N}$) concentrations. **(b)** Association between simultaneous satellite and normalized surface BC concentrations. The solid red line is the linear fit. The dashed grey line is the one-to-one line of $BC_{SAT}$ and $BC_{SUR-N}$.

**Figure 7.** Seasonal mean values of satellite-retrieved ($BC_{SAT}$) and surface-measured ($BC_{SUR}$ and $BC_{SUR-N}$) BC concentrations at different ARFINET sites (shown with respect to their latitudes) of India. The absolute differences between $BC_{SAT}$ and $BC_{SUR-N}$ are also shown. Panel **(a)** shows the seasonal values of $BC_{SAT}$, $BC_{SUR}$, $BC_{SUR-N}$, and $|(BC_{SAT} - BC_{SUR-N})|$ around each of the observational sites during December–January–February (DJF). The same parameters are shown in **(b)** for March–April–May (MAM) and in **(c)** for June–July–August (JJA). The letters in the histograms represent the names of individual stations (details in Table S1). Simultaneous data available for inter-comparison are the most in DJF (17 stations) and least in JJA (9 stations).

overpass time) day-to-day values of satellite-retrieved (1 km column-averaged) BC. The PBL height information is obtained from ERA5 (Hersbach et al., 2020). A similar methodology has been reported by Bao et al. (2019).

5   The frequency distributions of the absolute differences between the two datasets are depicted in Fig. 6a, which indicate good agreement between the simultaneous satellite-retrieved BC ($BC_{SAT}$) and normalized surface-measured BC ($BC_{SUR-N}$) concentrations. Approximately 60 % of $BC_{SAT}$
10  is comparable (absolute difference $< 2\,\mu g\,m^{-3}$) to $BC_{SUR-N}$ during all periods of DJF, MAM, and JJA. As depicted in Fig. 6b, correlation between the two datasets having absolute difference $< 2\,\mu g\,m^{-3}$ is the highest for MAM ($R \sim 0.76$), followed by DJF ($R \sim 0.73$) and JJA ($R \sim 0.61$).

15  The absolute differences between the two datasets are smaller (Fig. 7) at the PI locations where BC concentrations and seasonal variation are also lower than the northern Indian locations (seasonal mean values of surface-measured BC at each location are shown by the histograms). It is further evi-
20  dent from Fig. 7 that the absolute difference between $BC_{SAT}$ and $BC_{SUR-N}$ is markedly less than that between $BC_{SAT}$ and $BC_{SUR}$ at several locations of PI and northern India, especially during MAM and JJA. During winter, even though the

abundance of BC is confined near to the surface because of the shallow PBL condition, the noontime PBL is greatly ex-  25
tended (close to or beyond 1 km) over most of the Indian locations (the spatiotemporal variation in PBL height is shown in Fig. S6). Consequently, $BC_{SUR-N}$ follows the same general trend as that of the $BC_{SUR}$, indicating that noontime surface-measured BC concentrations during winter are sim-  30
ilar to the 1 km column-averaged BC. During MAM, the locations with PBL heights extended above 1 km are found to show a better association of $BC_{SAT}$ with $BC_{SUR-N}$ than that of $BC_{SAT}$ with $BC_{SUR}$. In JJA, the PBL height is found to be strongly region specific. At some locations, the PBL height is  35
much greater than 1 km (e.g., Chennai – CHN and Kadappa – KDP), whereas some other locations show the opposite pattern (i.e., Thiruvananthapuram – TVM, PBL height less than 1 km). The locations with PBL heights of less than 1 km are found to show lower absolute difference between $BC_{SAT}$  40
and $BC_{SUR-N}$ than between $BC_{SAT}$ and $BC_{SUR}$. However, it is also noteworthy that the simultaneous data of satellite-

retrieved and surface-measured BC are less in JJA than in DJF or MAM. Overall, it can be observed that, in most locations, the absolute difference between $BC_{SAT}$ and $BC_{SUR-N}$ is less than that between $BC_{SAT}$ and $BC_{SUR}$. This finding leads to better correlation between $BC_{SAT}$ and $BC_{SUR-N}$, especially during JJA, for which the correlation between $BC_{SAT}$ and $BC_{SUR-N}$ is much better ($R \sim 0.61$) than that between $BC_{SAT}$ and $BC_{SUR}$ ($R \sim 0.38$).

The northern part of India experiences significant seasonal changes in terms of incoming ground-reaching solar radiation, with intense radiation during pre-monsoon and monsoon months (Soni et al., 2012; Subba et al., 2022). This leads to significant seasonality in the PBL, which controls the vertical dispersion and therefore the near-surface loading (reduction) of aerosols. Based on airborne in situ measurements, Nair et al. (2016) have shown large seasonality (variation from winter to the pre-monsoon) in the vertical profile of aerosol absorption coefficients over the IGP and western India. Similarly, Brooks et al. (2019) reported a nearly uniform distribution of BC through the vertical profile over NW India, IGP, and the outflow region of IGP during the monsoon.

Apart from seasonality, BC over continental locations with low altitude above mean sea level shows significant diurnal variation with daytime lows and nighttime highs, with a sharp peak after the sunrise. Increased convective activity during daytime produces a deeper and more turbulent boundary layer and a faster dispersion of aerosols, resulting in decreased concentration near the surface. Several recent studies have described prominent effects of the PBL on the diurnal variation in BC, the amplitude of which varies considerably across the country, especially during winter (Babu et al., 2002; Beegum et al., 2009; Pathak et al., 2010; Gogoi et al., 2013, 2014; Kompalli et al., 2014; Prasad et al., 2018). In addition to variation in atmospheric mixing and vertical dispersion of BC, the accurate estimation of surface properties is another important parameter affecting satellite retrieval. Better estimates of surface properties during DJF and MAM might be the reason for improved correlation between satellite retrievals and surface BC concentrations, whereas the adverse atmospheric (hazy or cloudy) and land surface (wetter soils) conditions might affect the ability to estimate fine-mode aerosol concentrations during JJA.

### 3.1.3 Uncertainty of switching columnar concentrations to near the ground

With a view to understanding the uncertainty arising from the consideration of uniform distribution of BC within the PBL, the vertical profiles of BC obtained during two distinct periods of winter (December) and spring (May) over two distinct geographic locations of central (Hyderabad – HYD) and northern (Lucknow – LKN) India are considered based on data collected on board an instrumented aircraft as part of the Regional Aerosol Warming Experiment – RAWEX (Babu et

al., 2016; Gogoi et al., 2019). Because the vertical distribution of BC is not uniform in the real scenario, uncertainty arises from the estimated column BC amount from surface BC measurements and also from the derivation of BC from satellite-based measurements, which also assumes a uniform vertical distribution of BC within the well-mixed boundary layer. Figure S7 clearly illustrates that the vertical profiles of BC possess significant seasonality, in addition to their spatial variability. Up to the ceiling height of 1 km, average BC concentrations within this column appear to vary as high as 28 % (HYD) to 58 % (LKN) from that of the surface BC concentrations in winter. Compared to this, the columnar variability in spring is less (32 %) at LKN. Similarly, the columnar distribution of BC at HYD continued to show a sharp reduction with height to 1 km altitude but with subsequent enhancement in BC concentrations at greater heights. Based on the Model for Ozone and Related chemical Tracers version 4 (MOZART-4) simulation studies, Bao et al. (2019) have also reported that BC above the PBL contributes by 5 %–80 % to the column concentrations, even though the distribution of BC within the PBL is nearly uniform.

### 3.2 Soot volume fraction, SSA, and BC column optical depth

The soot volume fraction (SVF) or the volume fraction of BC in fine-mode particles is an important parameter that can reflect the relative dominance of soot in the fine-mode aerosol load in the column. Accurate estimates of SVF are necessary for the quantification of the radiative effects of BC (Wang et al., 2016). For this study, an internal mixture of fine-mode aerosols is adapted to represent aerosol light absorption by soot in fine-mode particles. The spatial distribution of the SVF at different months of winter, pre-monsoon, and monsoon seasons (as shown in Fig. S8) shows that the ratio of soot in the entire aerosol mixture is as high as 5 % over the IGP and northeastern parts of India. These values are similar to the mass fractions of BC reported by Gogoi et al. (2020) over the western, central, and eastern parts of the IGP based on airborne in situ measurements. Earlier in situ observations suggest that the values of SVF estimated from satellite-based observations can capture the broad regional features of columnar amounts of soot in fine-mode particle concentrations. Based on sensitivity studies, Hashimoto and Nakajima (2017) have reported that the detection of an absorption by soot and dust particles is less uncertain over a highly reflective surface and that the absorption is spectrally more sensitive to measurements of radiation at 380 nm of CAI-2 bands.

The monthly mean regional maps of SSA (at 546 nm) are shown in Fig. S9. The figure shows very large spatiotemporal variation, with values of SSA $< 0.92$ over most parts of the Indian region in December and January. In December, pockets of lower SSA (as low as 0.8) are observed over the western IGP, the Himalayan foothills, the NEI, and cen-

tral India. The values of SSA over the IGP remain low until March and April, which also depict low values ($\sim 0.8$) in its western part. It is evident from these observations that satellite-based retrieval of SSA from CAI-2 observations can quantify the spatiotemporal distribution of SSA, as found in several in situ measurements. Using aircraft measurements, Babu et al. (2016) reported the values of SSA between 0.86 and 0.94 over different western Indian and IGP locations during the pre-monsoon (April–May) period. The values of SSA in our study also show close agreement with those reported by Babu et al. (2016). In another study by Vaishya et al. (2018), the values of SSA reportedly decrease considerably over the Himalayan foothills, the IGP regions, and central India in the pre-monsoon compared to winter, whereas peninsular India and adjoining oceanic regions show an increase. Just before the monsoon onset, Vaishya et al. (2018) reported a decreasing gradient in SSA from the west to the east of the IGP ($\sim 0.84$ at the western IGP, 0.73 at the central IGP, and 0.79 at the eastern IGP, all at 530 nm). Over the oceanic regions, the values of SSA are generally high ($> 0.95$) and are comparable to the surface values reported over the entire Bay of Bengal (BoB) ($\sim 0.93$ during March–April) by Nair et al. (2008) and the Arabian sea ($\sim 0.9$ in March) by Jayaraman et al. (2001).

In contrast to the points raised above, the spatial distribution of SSA in our study was found to be different from that of the SSA derived from the Ozone Monitoring Instrument (OMI) on board the Aura satellite. The monthly maps of the regional distribution of SSA (at 550 nm) from OMI (level-3 daily 1° lat–long global gridded product OMAERUVd, https://disc.gsfc.nasa.gov/datasets/OMAERUV_003/summary, last access: 4 July 2023) are shown in Fig. S10. The difference between the regional distribution of SSA from CAI-2 and OMI is higher during DJF than during the other months. During DJF, CAI-2 retrievals show lower values of SSA over the Indian mainland compared to the OMAERUVd SSA. During JJA, the spatial patterns of SSA were similar in both CAI-2 and OMAERUVd retrievals.

Similarly to SVF and SSA, marked regional and seasonal differences in BC column optical depths ($BC_{AOD}$) are found (Fig. 8), with values from as low as 0.001 to as high as 0.1. During pre-monsoon months, $BC_{AOD}$ over the IGP shows a gradual decline during March–May; the pattern is the opposite over the NEI. Also, $BC_{AOD}$ shows pockets of high values over NEI in May. Increases in total columnar AOD over the IGP during March–May (peaks in June) were also reported by earlier investigators (Gautam et al., 2009, 2010) with an opposite trend (peak in March) over the NEI (Pathak et al., 2016). The higher $BC_{AOD}$ seen during December–April is indicative of the large amount of BC in the PBL during winter, both over the IGP (Singh et al., 2014; Vaishya et al., 2017) and NEI (Pathak et al., 2010; Guha et al., 2015) and its redistribution in the vertical column in the spring. This large amount of BC is modulated further by the occurrence

of seasonal fires over Southeast Asia, which start appearing in December and increase in spatial extent and magnitude over time, reaching a peak during March–May (Sahu et al., 2021).

## 3.3 Global distribution of BC from satellite retrievals

Considering the fair association between the satellite-retrieved and surface observations of BC over the Indian region, the global distribution of BC is examined for the different months of winter, the pre-monsoon, and the monsoon, as shown respectively in Figs. 9, 10, and 11. The global distribution of FRP (in MW) as shown in Figs. S11, S12, and S13 and the type of fire (presumed as vegetation fire, active volcano, static land shore and offshore; shown respectively in Figs. S14, S15, and S16) are also examined. This study uses only daytime FRP with a confidence level above 80 % (high confidence; Giglio et al., 2020).

Several typical hotspots of BC are observed throughout the year across the globe. They vary in magnitude, including many regions of South America, Africa, India, and China, with several of them coinciding with biomass burning activities. Significantly enhanced values of BC are also found for western Canada and the USA, over Europe, and Russia because of large fires occurring mainly during April–August. As shown in FRP maps in Figs. S11–S13, the fire activity increases in March over Southeast Asia, northeastern China, and some parts of southern and southeastern China. This pattern continues through May. For northern latitudes, the fire season begins in April–May. During the summer (JJA) season, the large-scale outbreak of forest fires in the boreal forests of North America (Macias Fauria and Johnson, 2008) and Russia (Cheremisin et al., 2022) is reported in the literature. In central Siberia, forest fires occur in April or at the beginning of May in southern areas and in June in northern areas (above 60° N latitude), with peak fire activity occurring in July (Kharuk et al., 2022). This tendency is readily apparent in the distribution of FRP (Figs. S11–S13). During 2019 and 2020, the fire activity in eastern Siberia was anomalously high (Xu et al. 2022), with higher total burned-out areas (Voronova et al., 2020). For the severe fire in 2019, the seasonal distribution (May–September) of fire frequency in the Siberian Arctic was bimodal, with modes of fire frequencies occurring in June and August (Kharuk et al., 2022). The smoke aerosols emitted continuously from these forest fires initially accumulated in southern Europe and Russia in May and spread up gradually to the northern latitudes in summer, resulting in the dispersion of the smoke plumes in the Arctic region. Apart from Siberia, during the summer (July–August) of 2019, anomalous wildfires occurred in Canada, Alaska, and Kazakhstan, as shown by the distribution of FRP and fire types. A similar pattern was also reported by Cheremisin et al. (2022). The fire activities over these regions start in April–May, contributing substantially to the aerosol emissions during spring. Junghenn Noyes

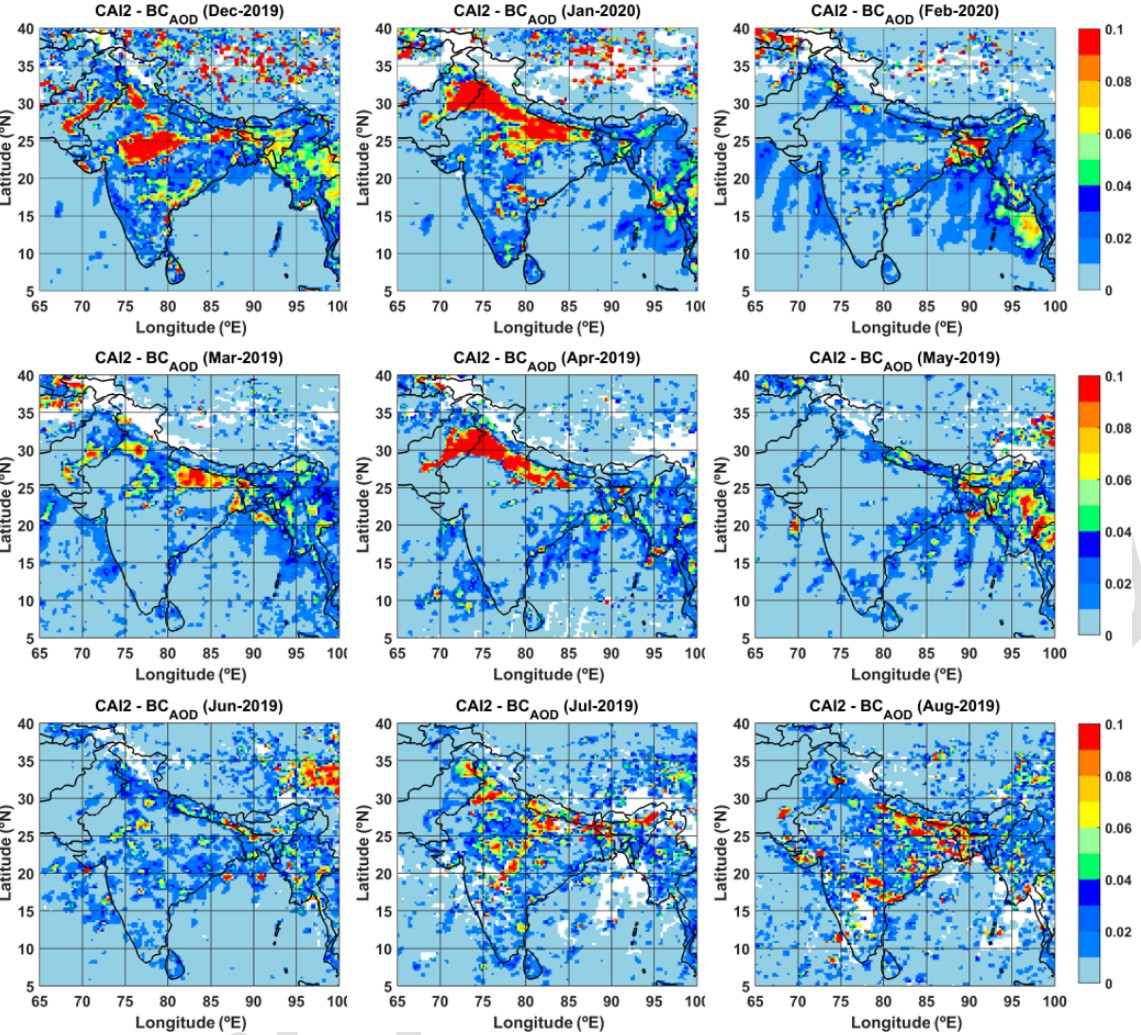

**Figure 8.** Regional distribution (0.25 × 0.25°) of monthly mean BC column optical depth (BC$_{AOD}$) over India during DJF, MAM, and JJA of 2019–2020.

et al. (2022) reported that Canadian and Alaskan wildfires inject higher amounts (percent) of plumes from forest and woody fires into the free troposphere in May.

In southern Africa, peak burning activity mainly takes place during July–September (Justice et al., 1996). However, the rainforest in central Africa, being the largest biomass burning region, shows a large increase in the magnitude of BC during both DJF and JJA, during which the biomass burning activities are also prominent. It is particularly interesting that some oceanic regions near the coast of western Africa also show higher values of BC during DJF and JJA. Some offshore fires are also seen to be contributing to the BC load on the west coast of Africa. In line with our observations, Barkley et al. (2019) reported the transport of African biomass burning aerosols to oceanic regions in the Southern Hemisphere. In another study based on GEOS-Chem-TOMAS global aerosol microphysics model simulations, Ramnarine et al. (2019) have reported the abundance

of organic aerosols and BC in the remote areas of the Southern Hemisphere downwind of biomass burning emissions from the Amazon in South America, the Congo in Africa, and some regions of the boreal forests in North America and Siberia. These observations clearly indicate that the spatiotemporal variation in BC across the globe is mostly coincident with the regions of intense biomass burning activities, whereas BC over some regions of southern Asia and China do not collocate with the biomass burning regions.

Satellite-based observations of global BC distribution examined for the present study are also found to be in line with those reported by Bond et al. (2004), showing the major areas of BC emissions over northern, central, and South America; Europe; Russia; the Middle East, the Pacific; Africa; China; and India. As reported from their study, substantial BC emissions from forest fire activity over South America and Africa are clearly reflected in satellite-retrieved BC data examined in our study. Similarly, higher BC load found over the re-

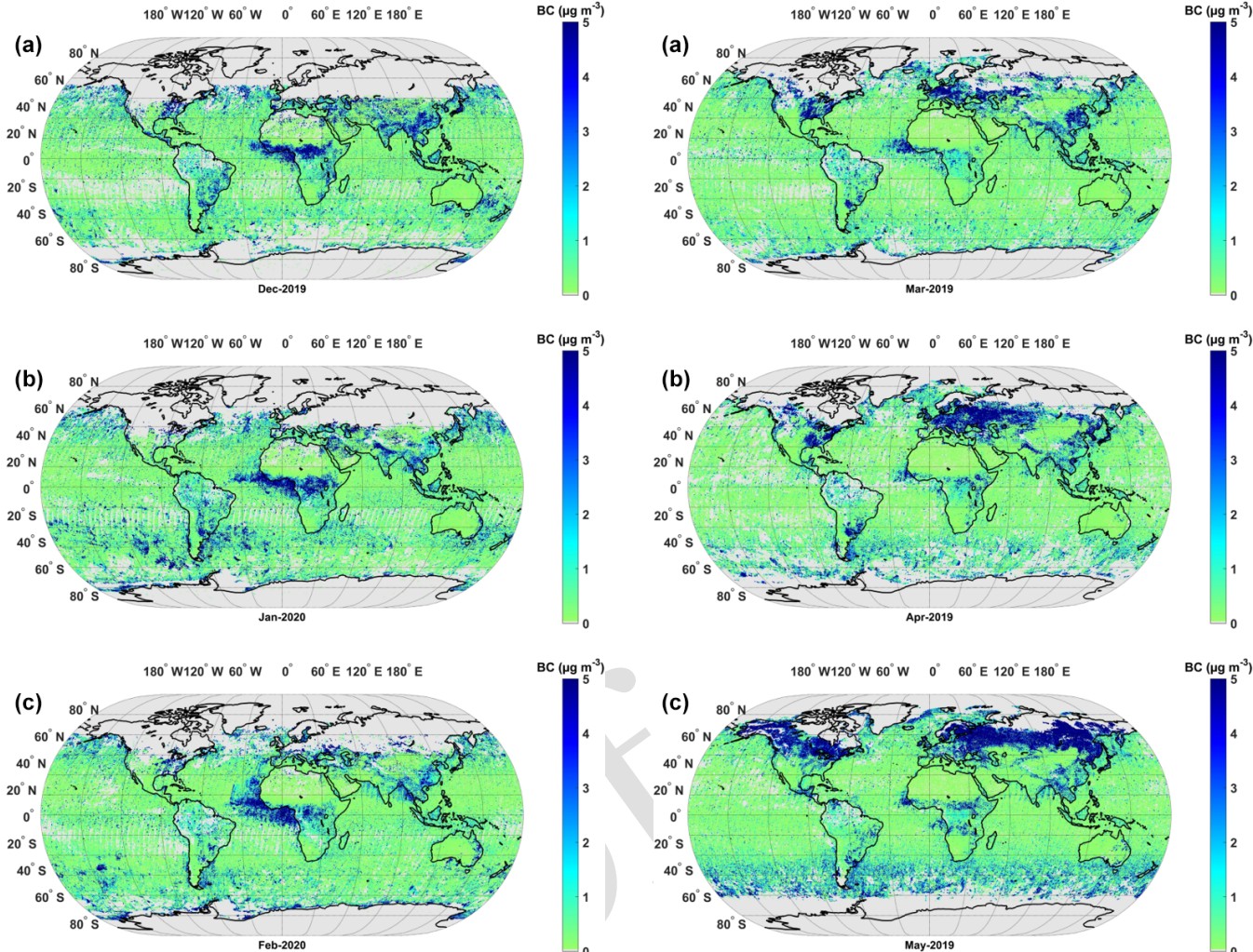

**Figure 9.** Global map of satellite-retrieved BC ($0.25 \times 0.25°$) during December (**a**) of 2019 and January (**b**) and February (**c**) of 2020.

**Figure 10.** Global map of satellite-retrieved BC ($0.25 \times 0.25°$) during March (**a**), April (**b**), and May (**c**) of 2019.

gions of Russia during the May–June period is coincident with open burning areas, as reported by Bond et al. (2004). Several reports (Dixon and Krankina, 1993; Leskinen et al., 2020, and the references therein) described that boreal forests and wild fires of Russia are crucially important in the context of the global carbon cycle, where large areas of burning Russian forest contribute to the net flux of carbon to the atmosphere.

## 4 Summary and conclusions

This study investigated the regional and global distribution of BC based on satellite retrievals. Extensive measurements of near-surface BC mass concentrations across a network of aerosol observatories (ARFINET) over the Indian region are used to evaluate the spatiotemporal distribution of BC retrieved from Cloud and Aerosol Imager-2 (CAI-2) on board the Greenhouse gases Observing Satellite-2 (GOSAT-2).

Regional distributions of BC from satellite retrieval (GOSAT-2 CAI-2) and surface measurements (ARFINET) during three distinct periods of December, January, and February (DJF); March, April, and May (MAM); and June, July, and August (JJA) showed good agreement between the two datasets over the Indian region. Especially during winter and pre-monsoon months, the satellite retrieval clearly identifies the regional hotspots of BC over India. Inter-comparison of satellite-retrieved BC with surface measurements revealed the absolute difference between the two datasets as $< 2 \, \mu g \, m^{-3}$ over 60 % of the observations in this study. Associations between the two datasets having an absolute difference $< 2 \, \mu g \, m^{-3}$ are the highest in MAM ($R \sim 0.76$), followed by DJF ($R \sim 0.73$) and JJA ($R \sim 0.61$).

The spatial distributions of the soot volume fraction (SVF) at different months of winter and pre-monsoon and monsoon

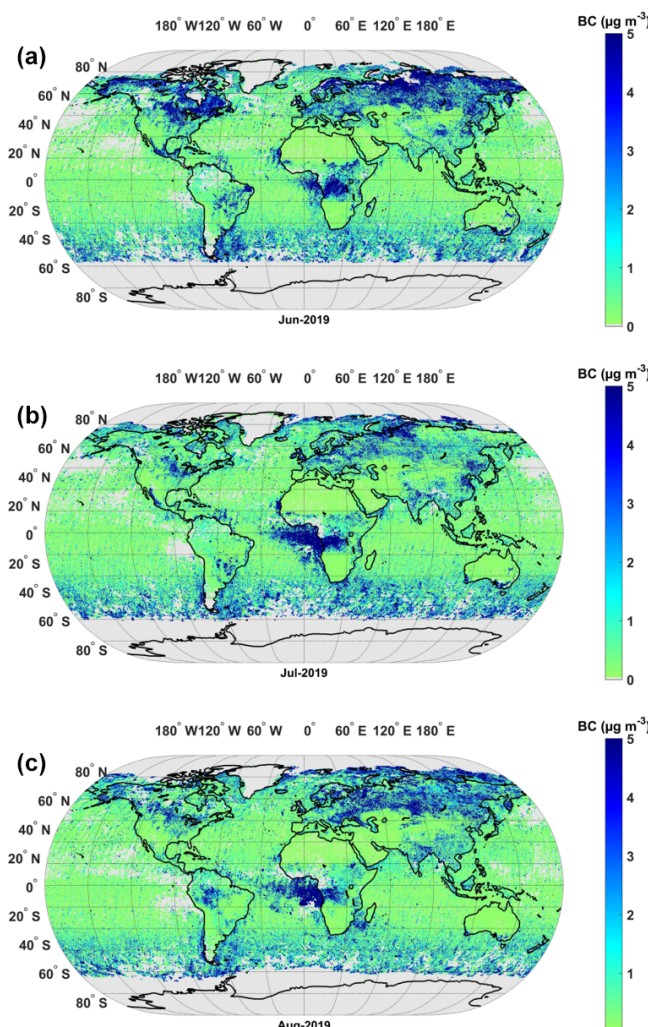

**Figure 11.** Global map of satellite-retrieved BC ($0.25 \times 0.25°$) during June (**a**), July (**b**), and August (**c**) of 2019.

seasons are similar to the spatial distribution of BC over the Indian region with the ratio of soot in the entire aerosol mixture of $> 5\%$ over the IGP and northeastern parts of India. The regional distribution of aerosol single-scattering albedo (SSA) shows values as low as 0.8 over the IGP and the northwestern part of India during winter and the pre-monsoon season. Similarly to SVF and SSA, marked regional and seasonal differences in BC column optical depths ($BC_{AOD}$) are apparent, with values ranging from as low as 0.001 to as high as 0.1. These observations are consistent with data reported from in situ measurements or other remote sensing platforms. All of these observations therefore suggest the applicability of the CAI-2 aerosol products.

Most of the spatiotemporal variation in BC across the globe occurs with intensive biomass burning activities, except for some regions of southern Asia and China. Enhanced values of BC are also found for western Canada and the USA, over Europe, Russia, and parts of China due to large fires burning mainly in summer. Across South America, Africa, India, and China, BC is generally higher throughout the year, not just during the biomass burning season.

**Data availability.** ARFINET data used for this study are available upon request from Surendran Nair Suresh Babu (s_sureshbabu@vssc.gov.in). GOSAT-2 TANSO-CAI-2 data used in this study are available at GOSAT-2 Product Archive: https://prdct.gosat-2.nies.go.jp/app/searchproduct/display (last access: 25 June 2023).

**Supplement.** The supplement related to this article is available online at: https://doi.org/10.5194/acp-23-1-2023-supplement.

**Author contributions.** This study was conceived by MMG and SSB in collaboration with RI and MH. Data processing and statistical analyses of the satellite and ground-based data were performed by MMG in consultation with SSB. All authors contributed to the manuscript conceptualization, editing, and review for submission. MMG drafted the initial manuscript with input from SSB. Regarding ground-based aerosol data, MMG and SSB were responsible for BC data from ARFINET, RI was the head of the science team of the GOSAT-2 project, and MH developed the inversion code. All authors read and approved the final paper.

**Competing interests.** The contact author has declared that none of the authors has any competing interests.

**Disclaimer.** Publisher's note: Copernicus Publications remains neutral with regard to jurisdictional claims in published maps and institutional affiliations.

**Acknowledgements.** This work was conducted as part of the ARFI project of ISRO-GBP. Mukunda Madhab Gogoi was the principal investigator of the Research Announcement on Greenhouse gases Observing SATellite Series (GOSAT RA). GOSAT-2/CAI-2 data were provided by JAXA/NIES/MOE. FRP (ftp://fuoco.geog.umd.edu, user name: fire, password: burnt, last access: 24 June 2023) data were obtained from the Moderate Resolution Imaging Spectroradiometer (MODIS). The global monthly fire location product (MCD14ML) was used for FRP. ERA-5 PBL data were obtained from ECMWF (https://cds.climate.copernicus.eu/cdsapp#!/dataset/reanalysis-era5-single-levels?tab=form, last access: 24 June 2023). The authors acknowledge Arun Gnanamony Sreekumar for his involvement in the processing of satellite and surface BC data. We also thank all the ARFINET investigators for the sustained efforts and support rendered over the years in maintaining the network and collecting data.

**Review statement.** This paper was edited by Zhanqing Li and reviewed by two anonymous referees.

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
