# Peer review of "Satellite (GOSAT-2 CAI-2) retrieval and surface (ARFINET) observations of Aerosol Black Carbon over India"

_Atmospheric Chemistry and Physics, 2022_

## Author Response (AR1)

**Satellite (GOSAT-2 CAI-2) retrieval and surface (ARFINET) observations of Aerosol Black Carbon over India**

Mukunda M. Gogoi et al., https://doi.org/10.5194/acp-2022-555

**Response to Referee #1**

The authors use extensive measurements to retrieve BC mass concentrations and compare them with satellite-retrieved BC mass concentrations. This paper contributes to better understanding of spatial and temporal distributions of BC concentrations over the Indian regions. I believe the paper should be considered for publication only after addressing the concerned expressed below.

**We appreciate the summary evaluation of the reviewer and agree to the observations. Following the valuable comments and fruitful suggestions for improving the quality of the manuscript, we have revised it incorporating the review comments of all the reviewers. Our point wise response to each of the comment is given below in bold letters, below the respective comments.**

Major Issues:

1. At first, it is difficult to read this paper because of unclear and unappropriated sentences. For readers to understand clearly, many sentences need to be polished with clear and concise structure including appropriate English.

**Response: We are sorry for the lack of clarity in certain areas. We have revised the manuscript and taken care of the clear and concise structures as suggested by the reviewer.**

2. Secondly, this paper mostly describes how much satellite and ground-based measurement agree or not. The authors need more descriptions about why they are different and what science is behind it. For example, you may investigate less consistency between satellite-retrieved and ground-based BC concentrations in JJA compared to DJF and MAM due to the cloud contamination or a deficit of data availability during monsoon.

**Response: We thank the reviewer for the valuable suggestion. Adequate descriptions are provided to explain the less consistency between satellite-retrieved and ground-based BC in the revised manuscript, as given below.**

**Line 318-334: "Based on the above observations it appears that the spatio-temporal distribution of BC as obtained from satellite retrievals show better consistency with the surface measured BC over the Indian region during DJF and MAM. As the rise in temperature caused by increased solar heating during MAM and JJA results in strong thermal convection over the Indian region (especially in the northern part), this leads to dilutions of near-surface aerosol concentrations. Depending upon the geographic position and local meteorological conditions, the strengths of convections vary from one location to the other. As the satellite retrieve BC is 1-km column average BC concentrations, the variation in the vertical distribution of BC may lead to variable associations between satellite-retrieved and surface measured BC concentrations at distinct geographic locations of India. More details on these aspects are discussed in the subsequent sections. Apart from the vertical heterogeneity, the various other**

factors that may lead to discrepancy in the satellite retrieval of BC include the bias caused by the cloud-screening algorithm, especially during JJA when the cloud cover over the Indian region is extensive. Moreover, CLAUDIA3 is unable to detect optically thin clouds. Lack of accurate detection of cloud shadow also can cause overestimation in the retrieve values of aerosol parameters from CAI-2 measurements. Since the revisiting time of CAI-2 is long (6 days), the minimum reflection criterion based on the consideration of 2 months data (one month prior and after the measurement days) can lead to large uncertainty in cloud-shadow detection, hence the accurate estimation of minimum surface reflectance. Subsequently, these errors can propagate and add uncertainty in the accurate estimation of aerosol parameters from CAI-2 measurements."

*Minor Issues and specific comments*:

Page1 L11: Is the acronym of ARFINET correctly located and explained? Aerosol Radiation Forcing over India NETwork (ARFINET)

**Response: Yes, Aerosol Radiation Forcing over India NETwork (ARFINET) of aerosols observatories is clearly mentioned.**

P1 L12: revealed -> reveals

**Response: Complied with**

P1 L17: that of other in-situ -> those of other in-situ

**Response: Complied with**

P1 L18: satellite retrieval shows -> satellite retrievals show

**Response: Complied with**

P1 L33: However, the very challenging task is to accurately retrieve -> However, it is challenging is to accurately retrieve

**Response: Complied with**

P2 L42: have not addressed so far -> have not been addressed so far

**Response: In the revised manuscript, the sentence is modified as**

**"… the retrieval of BC from satellite-based radiation measurement is very limited."**

P2 L50: 10-sepctral bands -> 10-spectral bands

**Response: Complied with**

P2 L50-51: use full word and abbreviation of UV, VIS, and NIR

**Response: Complied with**

P2 L59: to develop periodic and accurate estimates of aerosol radiative forcing over India and assess their impacts on regional and global climate, taking into account their heterogeneous properties in space, time and spectral domains -> to develop periodic and accurate estimates of aerosol radiative forcing over India, assess their impacts on regional and global climate, and take into account their heterogeneous properties in space, time and spectral domains

**Response: Complied with. The sentence is modified as:**

**"In the ARFINET, the main objective of the measurements of various aerosol parameters (e.g., columnar aerosol optical depth, BC mass concentrations, etc.) is to characterize their heterogeneous properties in space, time and spectral domains, develop periodic and accurate estimates of aerosol radiative forcing over India, and assess their impacts on regional and global climate."**

P2 L71: remove etc.

**Response: Complied with**

P3 L95: the surface albedo is derived by performing a correction removing the influence of atmospheric molecular scattering (Rayleigh scattering) -> the surface albedo is derived by removing the influence of atmospheric molecular scattering (Rayleigh scattering)

**Response: This section is now modified in the revised manuscript as**

**"After cloud and cloud shadow correction, the influence of atmospheric molecular scattering (Rayleigh scattering) is corrected from the minimum reflectance data."**

P3 L98: single scattering and multiple scattering -> single- and multiple-scattering

**Response: Complied with**

P3 L101: inversion algorithm developed by Hashimoto and Nakajima (2017) is used. -> inversion algorithm (Hashimoto and Nakajima, 2017) is used.

**Response: Complied with**

P3 L108: This sentence should be clearly stated.

aerosol light absorption (or single scattering albedo - SSA) -> aerosol light scattering (or single scattering albedo - SSA)

If you want to describe aerosol light absorption, you can use co-albedo or 1-SSA instead of SSA.

**Response: Complied with. We have maintained the consistency with aerosol light scattering (or single scattering albedo - SSA).**

P4 L131: Does "several sensitivity studies" means studies you performed? If not, add references.

**Response: Sorry for the confusing statement. The reference "Hashimoto and Nakajima, 2017" is included.**

P4 L148: Detail about the aethalometer uncertainty and correction of raw-data is available in Gogoi et al., (2017). The overall uncertainty in BC mass measured by the Aethalometer is estimated at about 10%. -> The overall uncertainty in BC mass measured by the aethalometer is estimated at about 10% and more details are available in Gogoi et al., (2017).

**Response: Complied with**

P5 L161: MAE = 10 $m^2g^{-1}$ is used. -> MAE = 10 $m^2g^{-1}$ is assumed (add references).

**Response: Complied with. The following citation is included:**

**Kondo, Y., Sahu, L., Kuwata, M., Miyazaki, Y., Takegawa, N., Moteki, N., Imaru, J., Han, S., Nakayama, T., Oanh, N. T. K., Hu, M., Kim, Y. J., and Kita, K.: Stabilization of the Mass Absorption Cross Section of Black Carbon for Filter-Based Absorption Photometry by the use of a Heated Inlet, Aerosol Science and Technology, 43, 741-756, https://doi.org/10.1080/02786820902889879, 2009.**

P5 L167: You mentioned, "As the ambient BC in the atmosphere is mostly aged in nature"

It is a vague sentence for the reason of "a value of MAE = 10 $m^2g^{-1}$ is used" since BC is not aged in nature if it is just released from biomass burning.

**Response: We thank the reviewer for pointing out the vague statement. We have revised the sentence as**

**"For estimating $\sigma_{abs}$ for the columnar content of BC, a constant value of mass absorption efficiency, MAE = 10 $m^2 g^{-1}$ is assumed (Kondo et al., 2009)."**

P5 L175: winter, pre-monsoon, and monsoon respectively. -> winter, pre-monsoon, and monsoon, respectively.

**Response: Complied with**

P5 L177: You should add one more figure or add text into Fig. 1 to indicate each region like HIM, IGP, NEI, NWI and so on. Although it is written in the supplement, this straightforward figure would help readers to understand your figure better.

**Response: Complied with the suggestion. We have included the following figure in the revised manuscript, clearly showing the regions of HIM, IGP, NEI, NWI, CI, PI and IL.**

[Figure]

**Figure-1: The network of aerosols observatories over India, distributed in the Indo-Gangetic Plains (IGP); North-eastern India (NEI); North-western India (NWI); Himalayan, sub-Himalayan and foothills regions (HIM), Central India (CI), Peninsular India (PI) and Island Locations (IL). More details about the ground-based observational locations in the ARFINET are provided in Supplementary Table-T1.**

P5 L195-197: You mentioned satellite retrievals and surface observations of BC are more consistent in DJF and MAM than JJA. Is less consistency in JJA caused by cloud contamination during monsoon? Or how many data used for this analysis for each season? Are the number of data used during monsoon fewer compared to other seasons?

**Response: We are thankful to the reviewer for the valuable suggestion. The discussions regarding the associations/ discrepancy between satellite-retrieved and surface measured BC are modified in the revised manuscript. The following section is also added to highlight the possible causes of higher discrepancy in the spatio-temporal distribution of BC over the Indian region during JJA as compared to that during DJF and MAM.**

**Line 318-334: "Based on the above observations it appears that the spatio-temporal distribution of BC as obtained from satellite retrievals show better consistency with the surface measured BC over the Indian region during DJF and MAM. As the rise in temperature caused by increased solar heating during MAM and JJA results in strong thermal convection over the Indian region (especially in the northern part), this leads to dilutions of near-surface aerosol concentrations. Depending upon the geographic position and local meteorological conditions, the strengths of convections vary from one location to the other. As the satellite retrieve BC is 1-km column average BC concentrations, the variation in the vertical distribution of BC may lead to variable associations between satellite-retrieved and surface measured BC concentrations at distinct geographic locations of India. More details on these aspects are discussed in the subsequent sections. Apart from the vertical heterogeneity, the various other factors that may lead to discrepancy in the satellite retrieval of BC include the bias caused by the cloud-screening algorithm, especially during JJA when the cloud cover**

over the Indian region is extensive. Moreover, CLAUDIA3 is unable to detect optically thin clouds. Lack of accurate detection of cloud shadow also can cause overestimation in the retrieve values of aerosol parameters from CAI-2 measurements. Since the revisiting time of CAI-2 is long (6 days), the minimum reflection criterion based on the consideration of 2 months data (one month prior and after the measurement days) can lead to large uncertainty in cloud-shadow detection, hence the accurate estimation of minimum surface reflectance. Subsequently, these errors can propagate and add uncertainty in the accurate estimation of aerosol parameters from CAI-2 measurements."**

P6 L204: better associations -> better agreement

**Response: Complied with.**

P6 L205: the association between the two data sets -> the correlation between the two data sets

**Response: Complied with.**

P6 L205: Thus, despite satellite retrievals during winter and pre-monsoon months showing the regional hotspots of BC over India fairly well, there appears to be a lack of consistent associations between the two datasets in winter at some of the ARFINET observational sites. -> Thus, satellite retrievals and surface observations show good agreement at the regional hotspots of BC over India during wither and pre-monsoon months. However, there are a lack of consistency between the two datasets in winter.

**Response: Complied with. We have modified the sentence as:**

**"This indicates that even though satellite retrievals and surface observations show good agreement at the regional hotspots of BC over India during winter and pre-monsoon months, there is a lack of consistency between the two datasets in winter at some of the other ARFINET observational sites."**

P6 L208: The above observations point to -> The discrepancies between satellite retrievals and ground-based observations can be attributed to

**Response: Complied with.**

P6 L215: Do not use "Despite this" and use specific words

**Response: We have removed the term "Despite this"**

P6 L215: the satellite retrievals differ from surface measured BC -> the satellite-retrieved BC differ from surface-measured BC

**Response: Complied with.**

P6 L228: we will now examine simultaneous day-to-day values -> we examine simultaneous day-to-day concentrations

**Response: Complied with.**

P6 L223-L225: This sentence needs to be polished.

**Response: Complied with. The sentence is modified as:**

**Lines 359-365: "In general, the surface measurements of BC concentrations over the entire Indian region show a gradual decline from its highest values in DJF (2.54 ± 0.11 µg m-3) through MAM (2.06 ± 0.47) to its lowest value in JJA (1.11 ± 0.17 µg m-3). Similar to this, the 1-km column average satellite retrieved BC also show highest BC concentrations over the collocated locations of India during DJF and their gradual decline in MAM. However, the satellite retrieved BC are found to be higher in JJA than in MAM, as opposed to the pattern seen in the case of surface measured BC. These observations hint again the discrepancy between satellite retrievals and surface measured BC in JJA, while their absolute magnitudes and regional distributions are nearly consistent during DJF and MAM in most locations."**

P7 L239:  play important role -> play an important role

**Response: Complied with.**

P7 L242: Add references of ERA5

**Response: Complied with. The following reference is included.**

**"Hersbach H., Bell, B., Berrisford P. et al.: The ERA5 global reanalysis. Quarterly Journal of Royal Meteorological Society, 146, 1999–2049, https://doi.org/10.1002/qj.3803, 2020."**

P7 L246: in all three periods of DJF, MAM and JJA -> during all periods

**Response: Complied with.**

P7 L249: It has also been observed that absolute differences between the two data sets -> Absolute differences between the two data sets

**Response: Complied with.**

P7 L249: peninsular Indian locations -> PI

**Response: Complied with.**

P7 L251: It is further evident from the figure -> It is further evident from Fig. 6

**Response: Complied with.**

P7 L259: Provide the reason of the sentence "Especially, the association between the two data sets significantly improves in JJA."

**Response: We sincerely thank the reviewer for suggesting a very important point to include in the discussions. Accordingly, we have elaborated the discussion as given blow.**

**Line 389-405:** "During winter, even though the abundance of BC is confined near to the surface due to shallow PBL condition, the noon time PBL is much extended (close to or beyond 1-km) over most of the Indian locations (the spatio-temporal variability in PBL height is shown in supplementary Fig. S6). Thus, $BC_{SUR-N}$ follows the same general trend as the $BC_{SUR}$, indicating that noon-time surface measured BC concentrations during winter are similar to the 1-km column average BC. During MAM, the locations with PBL heights extended above 1-km are found to show good association of $BC_{SAT}$ with $BC_{SUR-N}$ than that of $BC_{SAT}$ with $BC_{SUR}$. In JJA, the height of PBL is found to be highly region specific. At some of the locations, the PBL is much above 1-km (e.g., CHN and KDP), while some other locations show the opposite pattern (i.e., TVM, PBL height below 1 km). The locations with PBL heights below 1-km are found to show lower absolute difference between $BC_{SAT}$ and $BC_{SUR-N}$ than that between $BC_{SAT}$ and $BC_{SUR}$. However, it is also to be noted that the simultaneous data of satellite-retrieved and surface measured BC are less in JJA as compared to DJF and MAM. Overall, it is observed that, in most of the locations, the absolute difference between $BC_{SAT}$ and $BC_{SUR-N}$ is lower than that between $BC_{SAT}$ and $BC_{SUR}$. This leads to better correlation between $BC_{SAT}$ and $BC_{SUR-N}$, especially during JJA where the correlation between $BC_{SAT}$ and $BC_{SUR-N}$ is much better (R ~ 0.61) than that between $BC_{SAT}$ and $BC_{SUR}$ (R ~ 0.38)."

[Figure]

**Figure 7: Seasonal mean values of satellite-retrieved ($BC_{SAT}$) and surface-measured ($BC_{SUR}$ and $BC_{SUR-N}$) BC concentrations at different ARFINET sites (shown with respect to their latitudes) of India. The absolute difference between $BC_{SAT}$ and $BC_{SUR-N}$ are also shown. The top panel shows the seasonal values of $BC_{SAT}$, $BC_{SUR}$, $BC_{SUR-N}$ and |($BC_{SAT}$ - $BC_{SUR-N}$)| around each of the observational sites during December-January-February (DJF). Same parameters are shown in the middle panel for March-April-May (MAM) and**

bottom panel for June-July-August (JJA). The letters in the histograms represent the names of individual stations (details in supplementary Table-T1). The simultaneous data available for inter-comparison are highest in DJF (17-stations) and least in JJA (9-stations).

P7 L261: add references

**Response: The following references are included in support of the seasonal changes in the incoming ground reaching solar radiation in the northern part of India.**

**Soni, V.K., Pandithurai, G., Pai, D.S.: Evaluation of long-term changes of solar radiation in India. International Jouirnal of Climatology, 32 (4), 540–551, https://doi.org/10.1002/joc.2294, 2012.**

**Subba, T., Gogoi, M. M., Moorthy, K. K., Bhuyan, P. K., Pathak, B., Guha, A., Srivastava, M. K., Vyas, B. M., Singh, K., Krishnan, J., Lakshmikumar, T. V. S., Babu, S. S.: Aerosol Radiative Effects over India from Direct Radiation Measurements and Model Estimates, Atmospheric Research, 276, 106254, https://doi.org/10.1016/j.atmosres.2022.106254, 2022.**

P7 L267: show -> shows

**Response: Complied with.**

P7 L276: more wet soils -> wetter soils

**Response: Complied with.**

P8 L288: The forgone observation -> The prior observation

**Response: Complied with.**

P8 L298: Based on in-situ vertical profiling of aerosol scattering and absorption properties on a research aircraft, Babu et al., (2016) have reported the values of SSA between 0.86 and 0.94 over different West Indian and IGP locations during the pre-monsoon (April-May) period. -> Babu et al., (2016) have reported the values of SSA between 0.86 and 0.94 over different West Indian and IGP locations during the pre-monsoon (April-May) period using aircraft measurements.

**Response: Complied with.**

P8 L305: Over the oceanic regions, the values of SSA are, in general, high -> Over the oceanic regions, the values of SSA are generally high

**Response: Complied with.**

P8 L310-311: Mar -> Mar.

Jun -> Jun.

You may use abbreviations of the months consistently: decide full name or abbreviations of the months.

**Response: Complied with. The consistency is maintained.**

P8 L315: during Mar/Apr/May -> during March to May

**Response: Complied with**

P9 L319: Figs. 8, 9 and 10 -> Figs. 8, 9 and 10, respectively

**Response: Complied with.**

P9 L322: day time FRP -> day-time FRP

**Response: Complied with.**

P9 L343: add references after "Several studies"

**Response: Complied with. The following references are included in the revised manuscript.**

**Dixon, R. K., Krankina, O. N.: Forest fires in Russia: carbon dioxide emissions to the atmosphere, Canadian Journal of Forest Research, 23, 700-705, 1993.**

**Leskinen, P., Lindner, M., Verkerk, P.J., Nabuurs, G.J., Van Brusselen, J., Kulikova, E., Hassegawa, M. and Lerink, B. (eds.).: Russian forests and climate change. What Science Can Tell Us 11. European Forest Institute, 2020.**

P9 L345: evaluate and validate -> evaluate

**Response: Complied with.**

P9 L349: remove unnecessary sentence (The main findings are as follows:)

**Response: Complied with.**

P9 L351: do not use "fairly". It sounds informal.

**Response: Complied with.**

P9 L354: for > 60% of the observations (for all the locations considered in this study) the absolute difference between the two data sets is < 2 μgm$^{-3}$. -> the absolute difference between the two data sets is less than 2 μgm$^{-3}$ for over 60% of the locations in this study.

**Response: Complied with.**

P10 L365: during times of biomass burning -> during the biomass burning season

**Response: Complied with.**

P22 L654: Need more description about the plot (e.g., upper, center, and bottom panels indicate what) in the caption

**Response: Complied with. The figure caption is modified as**

**"Figure 7: Seasonal mean values of satellite-retrieved (BC$_{SAT}$) and surface-measured BC (BC$_{SUR}$ and BC$_{SUR-N}$) BC concentrations at different ARFINET sites (shown with respect to their latitudes) of India. The absolute difference between BC$_{SAT}$ and BC$_{SUR-N}$**

**at different locations are also shown. The top panel shows the seasonal values of $BC_{SAT}$, $BC_{SUR}$, $BC_{SUR-N}$ and $|(BC_{SAT} - BC_{SUR-N})|$ around each of the observational sites during December-January-February (DJF). Same parameters are shown in the middle panel for March-April-May (MAM) and in the bottom panel for June-July-August (JJA). The letters in the histograms represent the names of individual stations (details in supplementary Table-T1). The simultaneous data available for inter-comparison are highest in DJF (17-stations) and least in JJA (9-stations)."**

P24-26: Need more description about the plot (e.g., upper, center, and bottom panels indicate what) in the caption

**Response: Complied with.**

-END-

**Satellite (GOSAT-2 CAI-2) retrieval and surface (ARFINET) observations of Aerosol Black Carbon over India**

**Mukunda M. Gogoi et al.,** https://doi.org/10.5194/acp-2022-555

**Response to Referee #2**

General opinion:

The authors proposed an algorithm to retrieve black carbon from GOSAT-2 CAI-2. The authors also incorporated evaluation and validation of the satellite retrievals across a network of aerosol observatories (ARFINET) over India and the findings are extended to comprehend the global BC features. Such model is in highly demand if it is proven to work effectively. However, I am more concerned about the validity of the algorithm itself because the authors did not provide enough details on the methods, equations, and uncertainties. This may prevent the readers from understanding their work. Some descriptions and discussions are sometime puzzling, and there are thus much more revisions need to be made carefully by the authors.

**We appreciate the summary evaluation of the reviewer. We have complied with the observations and revised the manuscript incorporating valuable comments by the reviewers. In the revised manuscript, we have given more emphasis on the algorithm description, including the various steps involved in the retrieval process. The validation and uncertainties involved in this retrieval method is also elaborated. Our point wise response to each of the comment is given below in bold letters, below the respective comments.**

Major Comments

1. Inadequate innovation of the MS based on the claim of Line 40-41 "the direct retrieval of BC from satellite-based radiation measurement have not addressed so far." This is really not true. Below are some articles published in recent years, proposed similar algorithm in other countries.

   Bao, F., Cheng, T., Li, Y., Gu, X., Guo, H., Wu, Y., Wang, Y., & Gao, J. (2019). Retrieval of black carbon aerosol surface concentration using satellite remote sensing observations. Remote Sensing of Environment, 226, 93-108.

   Bao, F., Li, Y., Cheng, T., Gao, J., & Yuan, S. (2020). Estimating the Columnar Concentrations of Black Carbon Aerosols in China Using MODIS Products. Environmental Science & Technology, 54, 11025-11036.

   Li, L., Che, H., Derimian, Y., Dubovik, O., Schuster, G.L., Chen, C., Li, Q., Wang, Y., Guo, B., & Zhang, X. (2020). Retrievals of fine mode light-absorbing carbonaceous aerosols from POLDER/PARASOL observations over East and South Asia. Remote Sensing of Environment, 247, 111913.

**Response: We thank the reviewer for suggesting to include relevant works in our manuscript. The following information is added.**

Lines 37-55: "Even though several new algorithms have been developed for aerosol retrieval over land (e.g., Multi-Angle Imaging Spectroradiometer (MISR) retrieval by Dinner et al., 1998; Dark Target method by Levy et al., 2007; Non-linear optimal estimation algorithm for retrieval of aerosol microphysical properties from SAGE II satellite observations in the volcanically unperturbed lower stratosphere by Wurl et al., (2010); Multi-Angle Implementation of Atmospheric Correction (MAIAC) by Lyapustin et al., 2011; Deep Blue aerosol retrieval algorithm by Hsu et al., 2013; UV method by Fukuda et al., 2013; Multi-Angle and Polarization Measurements of Radiations by Dubovik et al., 2011, 2014; GOCI Yonsei Aerosol Retrieval (YAER) algorithm by Choi et al., (2016); Multi-Wavelength and -Pixel Method (MWPM) by Hashimoto and Nakajima, 2017 etc.), the retrieval of BC from satellite-based radiation measurement is very limited. Based on Effective Medium Approximations and statistically optimized aerosol inversion algorithm, Bao et al., (2019) have reported the retrieval of the surface mass concentration of BC from PARASOL (Polarization and Anisotropy of Reflectance for Atmospheric Sciences Coupled with Observations from a LiDAR) measurements. In another paper by Bao et al., (2020), MODIS Aqua Level-1B observations (MYD021KM) at three visible-infrared channels (470, 660, and 2100 nm) are used to estimate the columnar concentrations of BC aerosols based on BC and non-BC Maxwell−Garnett effective medium approximation (MG-EMA). POLDER/PARASOL satellite observations are also used by Li et al., (2020) to retrieve BC and brown carbon (BrC) concentrations based on aerosol component approach of Li et al., (2019). Apart from satellite observations, there are also efforts to retrieve BC from ground based remote sensing data. Hara et al., (2018) have reported the retrieval of BC from multi-wavelength Mie-Raman lidar (MMRL) observations, based on the modified algorithm of Nishizawa et al., (2017). Ceolato et al., (2022) have reported a direct and remote technique to estimate the BC number and mass concentration from picosecond short-range elastic backscatter lidar observations."

2. Comprehensive literature review and rigorous discussion is required in the introduction. And some details about the satellite sensor and data should be removed from the introduction to make the introduction concise.

**Response: Complied with the suggestion. New discussions citing the following literatures are included in the revised manuscript. The details about the satellite sensors are also shifted to the methodology section.**

**Wurl, D., Grainger, R. G., McDonald, A. J., and Deshler, T.: Optimal estimation retrieval of aerosol microphysical properties from SAGE~II satellite observations in the volcanically unperturbed lower stratosphere, Atmos. Chem. Phys., 10, 4295–4317, https://doi.org/10.5194/acp-10-4295-2010, 2010.**

**Choi, M., Kim, J., Lee, J., Kim, M., Park, Y.-J., Jeong, U., Kim, W., Hong, H., Holben, B., Eck, T. F., Song, C. H., Lim, J.-H., and Song, C.-K.: GOCI Yonsei Aerosol Retrieval (YAER) algorithm and validation during the DRAGON-NE Asia 2012 campaign, Atmos. Meas. Tech., 9, 1377–1398, https://doi.org/10.5194/amt-9-1377-2016, 2016.**

**Bao, F., Cheng, T., Li, Y., Gu, X., Guo, H., Wu, Y., Wang, Y., and Gao, J.: Retrieval of black carbon aerosol surface concentration using satellite remote sensing observations. Remote Sensing of Environment, 226, 93-108, 2019.**

Bao, F., Li, Y., Cheng, T., Gao, J., and Yuan, S.: Estimating the Columnar Concentrations of Black Carbon Aerosols in China Using MODIS Products. Environmental Science & Technology, 54, 11025-11036, 2020.

Ceolato, R., Bedoya-Velásquez, A.E., Fossard, F. et al.: Black carbon aerosol number and mass concentration measurements by picosecond short-range elastic backscatter lidar. Scientific Report, 12, 8443, https://doi.org/10.1038/s41598-022-11954-7, 2022.

Hara, Y., Nishizawa, T., Sugimoto, N., Osada, K., Yumimoto, K., Uno, I., Kudo, R., Ishimoto, H.: Retrieval of Aerosol Components Using Multi-Wavelength Mie-Raman Lidar and Comparison with Ground Aerosol Sampling. Remote Sensing, 10(6):937. https://doi.org/10.3390/rs10060937, 2018.

Li, L., Che, H., Derimian, Y., Dubovik, O., Schuster, G.L., Chen, C., Li, Q., Wang, Y., Guo, B., & Zhang, X.: Retrievals of fine mode light-absorbing carbonaceous aerosols from POLDER/PARASOL observations over East and South Asia. Remote Sensing of Environment, 247, 111913, 2020.

Li, L., Dubovik, O., Derimian, Y., Schuster, G. L., Lapyonok, T., Litvinov, P., Ducos, F., Fuertes, D., Chen, C., Li, Z., Lopatin, A., Torres, B., and Che, H.: Retrieval of aerosol components directly from satellite and ground-based measurements, Atmos. Chem. Phys., 19, 13409–13443, https://doi.org/10.5194/acp-19-13409-2019, 2019.

Nishizawa, T., Sugimoto, N., Matsui, I., Shimizu, A., Hara, Y., Itsushi, U., Kim, S.-W.: Ground-based network observation using Mie–Raman lidars and multi-wavelength Raman lidars and algorithm to retrieve distributions of aerosol components. Journal of Quantitative Spectroscopy and Radiative Transfer, 188, 79–93, 2017.

3. The authors should give a clear description of their algorithm. In section 2.1 the authors seem to spend a lot of space to review some other scholar's algorithms, which is confusing for some cross-field. In addition, did the authors use official unpublished products? The authors mentioned that the algorithm cite an under-preparation version of CAI-2 L2 aerosol retrieval ATBD (L117). If an official unpublished product is used, then a detailed description of the algorithm is needed. If the MS focuses on the improvements to existing algorithms, the basis, formulas, and the updates in this paper should also be emphasized. These descriptions must be detailed and not misleading.

**Response: We are sorry for the lack of clarity in the description of the algorithm. In the revised version of the manuscript, we have clearly highlighted that the data products used in this study are official unpublished products. Theoretical details about the retrieval of aerosol products from Cloud and Aerosol Imager (CAI) is available in Hashimoto and Nakajima (2017). The CAI-2 data products used in this study is also retrieved using the same principle, which is now clearly mentioned and described in the revised manuscript. In addition, various other steps (e.g., cloud discrimination and atmospheric corrections) involved in the retrieval process are clearly elaborated. Proper citations are also made to support the theoretical basis, formulas and the uncertainties involved in the retrieval process.**

**Lines 85-170:**

[revised manuscript text omitted]

4. I have a few doubts about the algorithm itself. Does the minimum reflectance strategy of surface reflectance correction in this MS consistent with that described in lines 81-85? What is the role of NDVI in this decision? In addition, if this strategy is used, it should be explained in detail in the flowchart (Fig. S1), as using 'minimum' may lead to misunderstandings.

**Response: We are sorry for the lack of clarity in this section. In the revised manuscript, we have clearly mentioned about the use of minimum reflectance criterion for the detection of cloud shadows. The relevant ATBD is cited. The information of NDVI is used for cloud discrimination over vegetated areas. The flowchart of the approach is revised and detail descriptions are included in the revised manuscript as given below.**

**Lines 92-108: "The cloud detection algorithm (Ishida et al. 2009, 2018) uses reflectance (at the top of atmosphere) of these bands for detecting clouds from 11 recurrences (one month before and after the observation date) (GOSAT-2 TANSO-CAI-2 L2 Cloud Discrimination Processing ATBD). A flow-chart of the Cloud and Aerosol Unbiased Decision Intellectual Algorithm (CLAUDIA3; Ishida et al., 2018; Oishi et al., 2017) employed for cloud-screening of GOSA-2 CAI-2 data is given in Supplementary Figure-S2. CLOUDIA3 is designed to automatically find the optimized boundary between clear and cloudy areas based on a supervised pattern recognition which uses support vector machines (SVM; Oishi et al., 2017). Before using the radiance (L1B) data in CLAUDIA3, a pre-processing is done to discriminate day and night, saturation flag, missing flag,**

polar region, water and land areas and sun-glint area for water area except Polar Regions. Following this, solar reflection properties by clouds and ground surface are examined, which includes: (i) solar reflectance and reflectance ratio in the VIS and SWIR regions, (ii) wavelength dependence of reflectance in the VIS and NIR region, (iii) NDVI test for cloud discrimination over vegetated areas, and (iv) reflectance ratio between NIR and SWIR bands for cloud discrimination over desert areas (details in Cloud Discrimination Processing ATBD). Subsequently, this information is used in the CLOUDIA3 algorithm, which performs the cloud discrimination by Support Vector Machine (SVM; Ishida et al., 2018) in order to objectively determine thresholds using multivariate analysis. SVM is one of the supervised pattern recognition methods, which first determines a decision function (called separating hyperplane) that defines clear or cloudy conditions according to the features of training samples (support vectors) in combination with a decision function."

5. In addition, the authors mentioned an internal mixing model to describe the proportion of BC in the aerosol. But it is not clear which internal mixing model are used. It is necessary to state and state the formula. How is the change in absorption of BC at different wavelengths considered? How is the absorption of other non-BC particles considered? The author defines: $f_{bc} = V_{soot}/V_{fine}$. It is also necessary to discuss the reasonableness of ignoring coarse particle aerosols. As far as I know, the spectral absorption of mixing aerosol is greatly influenced by some coarse particle (like DUST), which also show significant absorption in the near UV spectrum. These seemingly unreasonable assumptions can also have a very huge impact on later application studies.

**Response: We sincerely thank the reviewer for suggesting many important points to include in our discussions. Following this, we have elaborated the discussion on the estimation of soot volume fraction (SVF) as detailed below:**

**Lines 187-202: "For the estimation $f_{BC}$, an internal mixture of fine-mode aerosols (composed of 75% sulfuric acid and soot; mode radius ~ 0.175 μm and dispersion of the lognormal volume size distribution ~ 0.8) is considered and the volume fraction of soot particles (indicated as soot volume fraction, SF) is considered representative of aerosol light absorption by the fine-mode particles. Thus, $f_{BC} = V_{soot}/V_{fine}$, where $V_{soot}$ is the soot volume in the fine mode only. In the beginning, a-priori value of soot is assumed as 0.01 and the retrieval parameter 'u' is investigated based on its' a-priori state '$u_a$'. Several a-priori values around the true-states '$u_t$' are considered in the experiment; such as $u_t \pm 1.0u_t$ for $AOT_{500fine}$, $AOT_{500coarse}$, and SF, and $u_t \pm 0.01u_t$ for surface reflectance. The a-priori values of $AOD_{500fine}$ and $AOD_{500coarse}$ are considered as 0.2. The iteration in the solution search is stopped when the threshold is < 0.02.**

**In this simple approximation, various other mixing states of aerosols such as half internal and half external, core shell, and aggregated ones (Hashimoto et al., 2017 and references therein) are ignored. Thus, SF should be regarded as an equivalent value of soot in the fine mode particles, where the absorption property of aerosol is attributed only to the BC particles in the fine mode regime. As the BC mass distribution shows a mode of 100 – 300 nm (Kompalli et al., 2021) having stronger absorption in the NIR region, the light absorption by BC is significant mostly in the fine mode regime. The light absorption by other light-absorbing aerosols such as brown carbon and dust (coarse particles) responds strongly to radiation perturbation in the near-UV region (Mahowald et al., 2013). For the wavelength dependence of light absorption by BC, the complex refractive index of soot particles (d'Almeida et al., 1991) is considered in the retrieval process. However, the aerosol light absorption in the coarse mode domain is**

**not considered in this assumption. The complex refractive indices used as aerosol models for CAI-2 aerosol retrieval is shown in the figure below:**

[Figure]

**Figure: Complex refractive indices used as aerosol models for CAI-2 aerosol retrieval: (Fine mode: Sulfate + Soot, Coarse mode: Dust (Yello sand) and Sea-salt). Real part (top) and Imaginary part (bottom).**

6. In the validation section I note that the authors assume a uniformly columnar distributed BC, using a simple equation for the columnar concentration and near-surface conversion, but the ideal conditions are quite different from the actual observations. I would like to see a more reasonable solution. If not, I would like to see more validation, such as SSA, BCAOD, which makes the accuracy of the product more intuitive.

**Response: We fully agree with the reviewer that the columnar distribution of BC is not uniform in the real scenario. In this context, the uncertainty arising out of the consideration of uniform columnar distribution of BC from that of real BC variation with height is discussed in the revised manuscript. Further, it is clearly mentioned in the revised manuscript that the vertical distribution of BC is considered uniform in the well mixed layer, both in the retrieval algorithm as well as in the conversion of near surface BC to column concentration.**

**Lines 425-439: "With a view to understanding the uncertainty arising out of the consideration of uniform distribution of BC within the PBL, the vertical profiles of BC obtained during two distinct periods of winter (December) and spring (May) over two**

distinct geographic locations (Hyderabad – HYD and Lucknow - LKN) of central and northern India are considered based on data collected on-board an instrumented aircraft as part of Regional Aerosol Warming Experiment - RAWEX (Babu et al., 2016; Gogoi et al., 2019). As the vertical distribution of BC is not uniform in the real scenario, the uncertainty arises in the estimated column BC amount from surface BC measurements as well as in the derivation of BC from satellite-based measurements, which also assumes uniform vertical distribution of BC within the well mixed boundary layer. The supplementary Fig-S7 clearly shows that the vertical profiles of BC possess significant seasonality, in addition to their spatial variability. Up to the ceiling height of 1 km, it appears that the average BC concentrations within this column vary as high as 28% (HYD) to 58% (LKN) from that of the surface BC concentrations in winter. Compared to this, columnar variability in spring is relatively less (32%) at LKN. On the other hand, columnar distribution of BC at HYD continued to show a sharp reduction with height till 1 km altitude, but with subsequent enhancement in BC concentrations at higher heights. Based on Model for Ozone and Related chemical Tracers, version 4 (MOZART-4) simulation studies, Bao et al., (2019) have also reported that BC above the PBL contributes by 5%-80% to the column concentrations, even though the distribution of BC within the PBL is nearly uniform."

[Figure]

**Supplementary Fig-S7: Vertical profiles of BC (right panels) during two distinct periods of winter (December) and spring (May) over Hyderabad (central India) and Lucknow (Indo-Gangetic Plains). The horizontal bars show the standard deviations of the mean. The foot prints of data acquisition along the flight tracks are also shown in the left panels.**

**Intercomparison of SSA:**

**Lines 462-476:**

The values of SSA in our study are also in close agreement with those reported by Babu et al., (2016). In another study by Vaishya et al., (2018), it is reported that there is a significant reduction in the SSA over the Himalayan foothills, the IGP regions and central India in pre-monsoon as compared to the winter season; while the peninsular India and adjoining oceanic regions show an increase. Just prior to the onset of monsoon, Vaishya et al., (2018) have also reported a decreasing gradient in SSA from the west to the east of IGP (~ 0.84 at west IGP, 0.73 at central IGP and 0.79 at eastern IGP; all at 530 nm). Over the oceanic regions, the values of SSA are generally high (> 0.95) and comparable to the surface values reported over the entire BoB (~ 0.93 during March-April) by Nair et al., (2008); Arabian sea (~ 0.9 in March) by Jayaraman et al., (2001).

In contrast to the above, the spatial distribution of SSA in our study is found to be different from that of the SSA derived from Ozone Monitoring Instrument (OMI) onboard Aura satellite. The monthly maps of the regional distribution of SSA (at 550 nm) from OMI (Level-3 daily 1 deg Lat/Lon global gridded product OMAERUVd; https://disc.gsfc.nasa.gov/datasets/OMAERUV_003/summary) are shown in Supplementary Fig. S10. The difference between the regional distribution of SSA from CAI-2 and OMI is higher during DJF, as compared to that during the other months. During DJF, CAI-2 retrievals show lower values of SSA over the Indian mainland as compared to the OMEAUVd SSA. During JJA, the spatial patterns of SSA are similar in both CAI-2 and OMEAUVd retrievals.

[Figure]

**Fig.S10: Regional map (monthly average) of aerosol single scattering albedo (SSA) at 550 nm during DJF, JJA and MAM from OMEAUVd.**

7. In the comparison of Satellite retrievals vs climatological surface BC concentrations, do Satellite retrievals convert to near-ground magnitudes? If so, we need to move equation 3 here, but if not, the metrics RMSE in the

validation needs to be removed, because they are two parameters with different magnitudes.

**Response: Thanks for the suggestion. The metrics RMSE in the validation of satellite retrievals with the climatological surface BC concentrations is removed from Table-T2 as satellite retrievals are not converted to ground-magnitude for this inter-comparison.**

8. The uncertainty analyses is missing in the MS. i.e., the uncertainty of the algorithm itself; The uncertainty of interpolation; The uncertainty of internal mixing; The uncertainty of switching columnar concentration to near ground; The uncertainty of ignoring coarse particle aerosols. The uncertainty analyses are very important for those who use the product in the future.

**Response: We are very much thankful to the reviewer for the valuable comment. Following details on uncertainty are included in the revised manuscript. Since, the core algorithm of retrieving AOD from CAI-2 measurements is based on Hashimoto et al., 2007, several inferences regarding the uncertainty and error analysis is cited from this article.**

**Uncertainty analysis of the algorithm:**

**(Lines 171-177): Uncertainty of AOD and SSA retrieval**

**The uncertainty in the retrieval of AOD using MWPM inversion algorithm over heterogeneous surfaces is found to be within ±0.062, ±0.048 and ±0.077 for $AOD500_{fine}$, $AOD500_{coarse}$ and $AOD500_{total}$ respectively (Hashimoto and Nakajima, 2017). These results are based on the comparison of AOD retrieval from CAI measurements of radiances with AOD data obtained from AERONET (Holben et al., 1998) and SKYNET (Nakajima et al., 2007). Comparison of the CAI-retrieved SSA (at 674 nm) with that of the AERONET observed values (SSA at 675 nm) revealed the retrieval accuracy of SSA within 0.05. Over the homogeneous surface, the random measurements error of the retrieval parameters is below 2%.**

**The uncertainty of internal mixing and the uncertainty of ignoring coarse particle**

**Lines 203-220: With a view to understanding the uncertainty of satellite received radiances due to different mixing states of aerosols with varying BC fractions, a sensitivity study is carried out using 6S radiative transfer code (Vermote et al., 1997). 6S code is widely used for the simulation of satellite reaching radiation for different combinations of sun-satellite geometry under various conditions of aerosol load in the atmosphere. The surface is considered as homogeneous Lambertian surface in the simulations. It is observed (Supplementary Figure-S3) that the sensitivity of BC-fraction (at 880 nm) to satellite reaching radiation is significantly improved under higher aerosol loadings (AOD > 0.5) as well as under higher surface reflectance conditions, while there is no marginal distinction between BC and non-BC conditions for AOD < 0.5. The sensitivity study also clearly indicates that the satellite reaching radiation for 1% BC in the aerosol mixture are affected by as low as 5% for variation in dust fraction from 1% to 10% during low aerosol loading conditions (AOD ~ 0.1). For higher BC fraction (~ 10%) in the aerosol mixture under heavy aerosol loading conditions (AOD ~ 2.0), the variation in dust fraction from 1% to 10% is found to change the apparent reflectance by ~10% for surface conditions of higher reflectance (~ 0.5), while the variability is much larger (~ 15%) for low surface reflectance conditions (~ 0.1). This exercise clearly indicates that the uncertainty in satellite retrieval of BC arising out of ignoring the**

contribution of dust in the aerosol mixture is less over dark surfaces when the aerosol load is low. Similarly, the retrieval uncertainty is lower over brighter surface when the aerosol load is high. Overall, it is to be noted that consideration of the accurate mixing state (internal and external) of aerosols is important for accurate computation of effective refractive index and size distribution of aerosols. Lesins et al., (2002) have reported that the optical properties of the internal mixture of BC and ammonium sulfate can differ by as high as 25% (for the dry case) and 50% (for the wet case) from that of its external mixture.

[Figure]

**Supplementary Figure-S3:** Variability of apparent reflectance of satellite observation at 0.880 μm wavelength with surface reflectance for different fractions of BC (1%, 5%, 10% and 20%), dust (1% and 10%) under different conditions of AOD (0.1, 0.5, 1.0, 2.0). The fraction of water-soluble species is kept constant (50%). The solar zenith and azimuth angles are 40° and 100°, and satellite viewing angle and azimuth angle are 45° and 50° respectively. The surface reflectance is considered for homogeneous Lambertian surface.

Within the above-mentioned uncertainties, the sensitivity study has shown that SF is underestimated under low aerosol loading conditions (AOD < 0.2) over highly-reflective surface. This is because the retrieval uncertainty of AOD is higher over the high-reflectance surface which leads to the overestimation of $AOD_{500fine}$. For higher aerosol loading condition ($AOT_{500total} > 0.4$), the MWPM algorithm simultaneously determines $AOT_{500fine}$, $AOT_{500coarse}$, and SF within error of ±0.06, ±0.05, and ±0.05 respectively.

**The uncertainty of switching columnar concentration to near ground**

**Lines 425-440**: "With a view to understanding the uncertainty arising out of the consideration of uniform distribution of BC within the PBL, the vertical profiles of BC obtained during two distinct periods of winter (December) and spring (May) over two distinct geographic locations (Hyderabad – HYD and Lucknow - LKN) of central and northern India are considered based on data collected on-board an instrumented aircraft as part of Regional Aerosol Warming Experiment - RAWEX (Babu et al., 2016; Gogoi et al., 2019). As the vertical distribution of BC is not uniform in the real scenario, the uncertainty arises in the estimated column BC amount from surface BC measurements as well as in the derivation of BC from satellite based measurements, which also assumes uniform vertical distribution of BC within the well mixed boundary layer. The supplementary Fig-S7 clearly shows that the vertical profiles of BC possess significant seasonality, in addition to their spatial variability. Up to the ceiling height of 1 km, it appears that the average BC concentrations within this column vary as high as 28% (HYD) to 58% (LKN) from that of the surface BC concentrations in winter. Compared to this, columnar variability in spring is relatively less (32%) at LKN. On the other hand, columnar distribution of BC at HYD continued to show a sharp reduction with height till 1 km altitude, but with subsequent enhancement in BC concentrations at higher heights. Based on Model for Ozone and Related chemical Tracers, version 4 (MOZART-4) simulation studies, Bao et al., (2019) have also reported that BC above the PBL contributes by 5%-80% to the column concentrations, even though the distribution of BC within the PBL is nearly uniform."

[Figure]

**Supplementary Fig-S7: Vertical profiles of BC during two distinct periods of winter (December) and spring (May) over Hyderabad (central India) and Lucknow (Indo-Gangetic Plains). The foot prints of the data acquisition along the flight tracks are also shown in the left panels. The horizontal bars show the standard deviations of the mean.**

9. How are SSA and FRP calculated in the section2-3.3? It is not reasonable to extrapolate Indian retrievals to global FRP without extended validation and uncertainty analyses, and it may be more convincing to state Indian only.

**Response: The retrieval of SSA and FRP is elaborated in the revised version of the manuscript as given below. Reviewer may also kindly note that the FRP used in this study is MODIS Collection 6 Active Fire Products (MCD14ML), which are extensively validated (e.g., Giglio et al., 2016) and used by many investigators to estimate the contribution of biomass burning to local and global carbon budgets. Considering this, we have also retained the global distribution of FRP in the revised manuscript, even though MODIS FRP has an uncertainty of ~ 26% at the 1 sigma level.**

**Response: Sorry for the typo error. The statistical parameters are corrected (for January) in Figure-S2 (Figure-S4 in the revised manuscript).**

-END-

---

## Author Response (AR2)

**Response to the Editor**

*While major improvements are noted, one reviewer still has some concerns. Besides, I'd question the high values of BC in Siberia in May (Fig. 10) when there are few wildfires in the region. If they are caused by fire smoke, a more in-depth investigation should be done to test the "hypothesis".*

**Response: We appreciate the comment from the editor and revised the manuscript as given below.**

**"Several typical hotspots of BC are observed throughout the year across the globe. They vary in magnitude, including many regions of South America, Africa, India, and China, with several of them coinciding with biomass-burning activities. Significantly enhanced values of BC are also found for western Canada and the USA, over Europe, and Russia, because of large fires occurring mainly during April - August. As shown in FRP maps in Figs. S11 - S13, the fire activity increases in March over Southeast Asia, north-eastern China, and some parts of southern and southeastern China. This pattern continues through May. For northern latitudes, the fire season begins in April - May. During the summer (JJA) season, the large-scale outbreak of forest fires in the boreal forests of North-America (Fauria and Johnson, 200.8) and Russia (Cheremisin et al., 2022) are reported in the literatures. In central Siberia, forest fires occur in April or at the beginning of May in southern areas, and in June in northern areas (above 60°N latitude), with peak fire activity occurring in July (Kharuk et al., 2022). This tendency is readily apparent in the distribution of FRP (Figs. S11-S13). During 2019 and 2020, the fire activity in eastern Siberia was anomalously high (Xu et al. 2022), with higher total burned-out areas (Voronova et al., 2020). For the severe fire in 2019, the seasonal distribution (May - September) of fire frequency in the Siberian Arctic was bimodal, with modes of fire frequencies occurring in June and August (Kharuk et al., 2022). The smoke aerosols emitted continuously from these forest fires initially accumulated in the southern Europe and Russia in May and spread up gradually to the northern latitudes in summer, resulting in the dispersion of the smoke plumes in the Arctic region. Apart from Siberia, during the summer (July - August) of 2019, anomalous wildfires occurred in Canada, Alaska, and Kazakhstan, as shown by the distribution of FRP and fire types. A similar pattern was also reported by Cheremisin et al. (2022). The fire activities over these regions start in April - May, contributing substantially to the aerosol emissions during spring. Noyes et al. (2022) reported that Canadian and Alaskan wildfires inject higher amounts (percent) of plumes from forest and woody fires in to the free-troposphere in May."**

**Response to Reviewer 1**

*Even though it is second version of review, there are still many in-appropriate English (typos and inconsistencies in caption number) which may distract readers from reading this paper.*

**Response: We thank the reviewer for highlighting the inconsistencies as well as inappropriate sentences. We have carefully revised the manuscript taking care of the English and rectified the inconsistencies.**

**Response to Reviewer-2**

*I appreciate the author's large amount of modification work. I think most of the comments have been well responded. I suggest author further **glorify the innovation of the algorithm based on the new references of satellite BC retrievals,** and to further improve the English writing before accepting it.*

**Response: We sincerely thank the reviewer for the valuable comments/ suggestions and for accepting our responses to the queries. In the revised version of the manuscript, we have highlighted the innovation of the algorithm (as given below) in comparison with the recent studies of satellite BC retrievals. The English correction is also made with the help of an professional expert.**

[revised manuscript text omitted]

-END